# Impact of blindness onset on the representation of sound categories in occipital and temporal cortices

**Stefania Mattioni[1,2]\*, Mohamed Rezk[1], Ceren Battal[1], Jyothirmayi Vadlamudi[1], Olivier Collignon[1,3,4,5]\***

[1]Institute for research in Psychology (IPSY) & Neuroscience (IoNS), Louvain Bionics, Crossmodal Perception and Plasticity Laboratory - University of Louvain (UCLouvain), Louvain-la-Neuve, Belgium; [2]Department of Brain and Cognition, KU Leuven, Leuven, Belgium; [3]Center for Mind/Brain Studies, University of Trento, Trento, Italy; [4]School of Health Sciences, HES-SO Valais-Wallis, Sion, Switzerland; [5]The Sense Innovation and Research Center, Lausanne and Sion, Sion, Switzerland

**\*For correspondence:**
stefania.mattioni@uclouvain.be (SM);
olivier.collignon@uclouvain.be (OC)

**Competing interest:** The authors declare that no competing interests exist.

**Abstract** The ventral occipito-temporal cortex (VOTC) reliably encodes auditory categories in people born blind using a representational structure partially similar to the one found in vision (Mattioni et al.,2020). Here, using a combination of uni- and multivoxel analyses applied to fMRI data, we extend our previous findings, comprehensively investigating how early and late acquired blindness impact on the cortical regions coding for the deprived and the remaining senses. First, we show enhanced univariate response to sounds in part of the occipital cortex of both blind groups that is concomitant to reduced auditory responses in temporal regions. We then reveal that the representation of the sound categories in the occipital and temporal regions is more similar in blind subjects compared to sighted subjects. What could drive this enhanced similarity? The multivoxel encoding of the 'human voice' category that we observed in the temporal cortex of all sighted and blind groups is enhanced in occipital regions in blind groups , suggesting that the representation of vocal information is more similar between the occipital and temporal regions in blind compared to sighted individuals. We additionally show that blindness does not affect the encoding of the acoustic properties of our sounds (e.g. pitch, harmonicity) in occipital and in temporal regions but instead selectively alter the categorical coding of the voice category itself. These results suggest a functionally congruent interplay between the reorganization of occipital and temporal regions following visual deprivation, across the lifespan.

## Editor's evaluation

The study interrogates the representational structure of sound categories in the temporal cortex of early- and late-onset blind people. This adds two novel dimensions to the author's previous focused on auditory categorical representation in the visual cortex of people with early blindness onset, and as such will be of interest to researchers studying brain reorganisation across life. The strength of the study is in its methodology, which provides compelling and robust evidence to support the study's main conclusions.

## Introduction

The occipital cortex of early blind (EB) individuals enhances its response to non-visual stimuli (*Neville and Bavelier, 2002*). For instance, it has been repetitively shown that sound processing triggers

enhanced occipital responses in EB people (*Van Ackeren et al., 2018*; *Bedny et al., 2011*; *Collignon et al., 2011*; *Dormal et al., 1981*; *Weeks et al., 2000*).

If occipital regions enhance their functional tuning to auditory information in EB, what is the impact of visual deprivation on temporal regions typically coding sounds? Contradictory results emerged from previous literature about the way intramodal plasticity expresses in early blindness. Several studies suggested that visual deprivation elicits enhanced response in the sensory cortices responsible for touch or audition (*Elbert et al., 2002*; *Gougoux et al., 2009*; *Manjunath et al., 1998*; *Naveen et al., 1998*; *Pascual-Leone and Torres, 1993*; *Rauschecker, 2002*; *Röder et al., 2002*). In contrast, some studies observed a decreased engagement of auditory or tactile sensory cortices during non-visual processing in EB individuals (*Bedny et al., 2015*; *Burton et al., 2002*; *Pietrini et al., 2004*; *Ricciardi et al., 2009*; *Stevens and Weaver, 2009*; *Striem-Amit et al., 2012*; *Wallmeier et al., 2015*). Those opposing results were, however, both interpreted as showing improved processing in the regions supporting the remaining senses in blind people: more activity means enhanced processing and less activity means lower resources needed to achieve the same process; so, both more and less mean better. In this fallacious interpretational context, the application of multivoxel pattern analysis (MVPA) methods to brain imaging data represents an opportunity to go beyond comparing mere activity level differences between groups by allowing a detailed characterization of the information contained within brain areas (*Berlot et al., 2020*; *Kriegeskorte et al., 2008b*). An intriguing possibility, yet to be directly tested, is that early visual deprivation triggers a redeployment mechanism that would reallocate part of the sensory processing typically implemented in the preserved senses (i.e. the temporal cortex for audition) to the occipital cortex deprived of its dominant visual input.

A few studies reported an increased representation of auditory stimuli in the occipital cortex concomitant to a decreased auditory representation in temporal regions in congenitally blind people (*Battal et al., 2021*; *Dormal et al., 2016*; *Jiang et al., 2016*, *van den Hurk et al., 2017*; *Vetter et al., 2020*). However, these studies did not focus on the link between intramodal and crossmodal reorganizations in blind individuals. For instance, we do not know, based on this literature, whether this increased/decreased representation is driven by similar or different features of the auditory stimuli in temporal and occipital regions. We have recently demonstrated that categorical membership is the main factor that predicts the representational structure of sounds in ventral occipito-temporal cortex (VOTC) in congenitally blind people (*Mattioni et al., 2020*), rather than lower-level acoustical attributes of sounds (i.e. pitch). Would the same categorical representation be the one that could be reorganized in the temporal cortex of these blind individuals? If true this would speak up for an interplay between the features that are reorganized in the temporal and occipital cortices of visually deprived people. Alternatively, the intramodal reorganization potentially observed in the temporal region of blind people might be driven by the acoustic properties of sounds, suggesting reorganization of independent auditory features (acoustic vs. categorical) in temporal and occipital regions. Representational similarity analyses (RSA) can reveal whether categorical vs. acoustic representation of the same set of sounds is encoded in a brain region (*Giordano et al., 2013*). Here, using RSA, we explore for the first time which features of the sounds (acoustic or categorical) are concomitantly reorganized in the temporal or occipital cortex of blind compared to sighted people.

Another unsolved question relates to how the onset of blindness impacts the organization of cortical regions coding for the preserved and deprived senses. We have recently suggested that the increased representation of sound categories in the VOTC of EB people could be an extension of

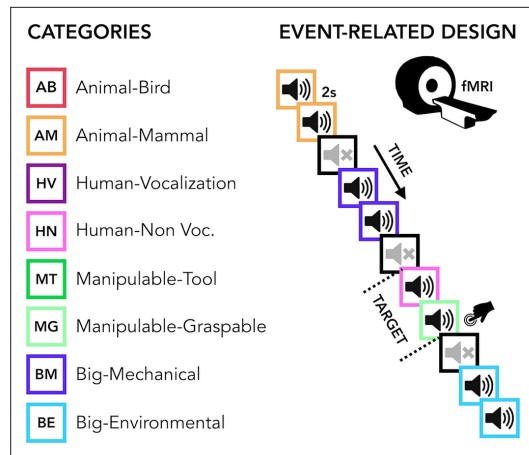

**Figure 1.** Experimental design. (Left) Categories of stimuli. (Right) Design of the fMRI experiment. (Right) Regions of interest (ROIs) selected from groups' contrasts. β-Values from each group and ROIs for every main category (animal, human, manipulable, big objects and places) are reported in the orange (temporal) and green (occipital) rectangles.

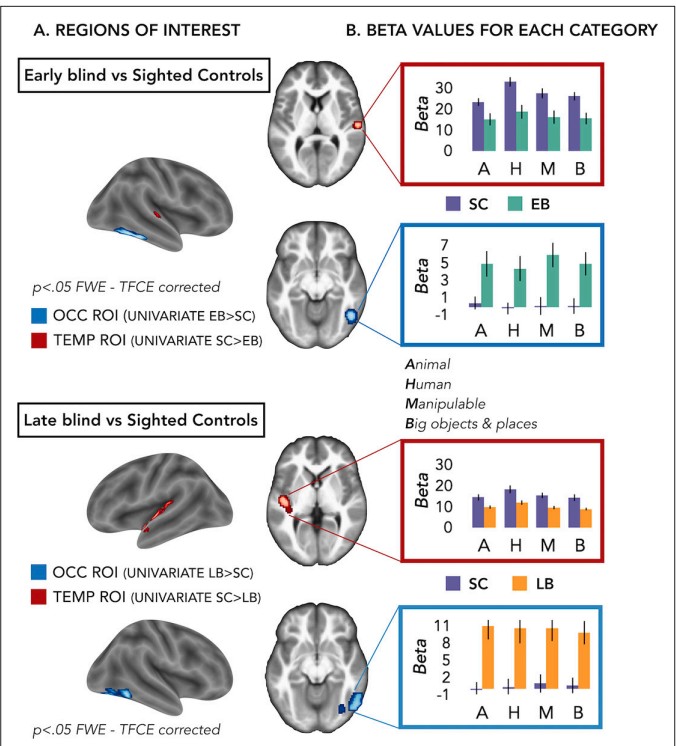

**Figure 2.** Regions of interest (ROIs).
(**A**) ROIs selected from groups' contrasts at the univariate level. Note that, for illustration purpose, we report here the groups univariate contrasts including all subjects, but to avoid circular analyses we actually created ad hoc ROIs using a leave-one-subject-out approach (i.e. for each subject we excluded himself/herself from the univariate contrast). (**B**) β-Values from each group and ROIs for every main category (animal, human, manipulable, big objects and places) are reported in the red (temporal) and blue (occipital) rectangles. Black bars show standard error (sample size: Early Blind=16; Late Blind=15; Sighted Controls=20).

The online version of this article includes the following figure supplement(s) for figure 2:

**Figure supplement 1.** Univariate whole brain analyses.

the intrinsic multisensory categorical organization of the VOTC, that is therefore partially independent from vision in sighted as well (*Mattioni et al., 2020*; see also *Amedi et al., 2002*; *Ricciardi and Pietrini, 2011*). According to this view, one should assume that late visual deprivation may extend the non-visual coding that is already implemented in the occipital cortex of sighted people. In contrast with this hypothesis, previous studies suggested that late acquired blindness triggers a reorganization of occipital region that is less functionally organized than the one observed in early blindness (*Bedny et al., 2012*; *Collignon et al., 2013*; *Kanjlia et al., 2009*), promoting the idea that crossmodal plasticity in late blindness is more stochastic and functionally epiphenomenal compared to the one observed in EB people.

The current study aimed to carry out a comprehensive uni- and multivariate characterization of how early and late acquired blindness impact the processing of sounds from various categories (humans, animals, manipulable objects and big objects or scenes, *Figure 1* ) in occipital and temporal regions.

## Results

### β's extraction

We defined our regions of interest (ROIs) based on group differences of the univariate results (*Figure 2* and *Figure 2—figure supplement 1*). However, in these univariate contrasts, we included the sounds from all the different categories. Is one of our four main categories (i.e. animal, human, manipulable objects, and big objects and places) driving these groups' differences (*Figure 1*)?

To address this point we extracted the β-values in each ROI from every subject for every main category (see *Supplementary file 3* and *Figure 2B*) and we looked if there was a significant interaction Group*Category.

For the EB/sighted control (SC) comparisons, we run two separate ANOVA, one in the occipital ROI and one in the temporal ROI.

In the occipital ROI (from the univariate contrast EB > SC), we observed a significant main effect of Group ($F_{(1,34)}$=11.91; p=0.001) while the main effect of Category ($F_{(3,102)}$=1.22; p=0.31) and the interaction Group*Category ($F_{(3,102)}$=0.76; p=0.52) were both non-significant.

In the temporal ROI (from the univariate contrast SC > EB), we observed a significant main effect of Group ($F_{(1,34)}$=8.23; p=0.007) and a significant main effect of Category ($F_{(3,102)}$=12.29; p<0.001), while the interaction Group*Category ($F_{(3,102)}$=1.93; p=0.13) was not significant. The post hoc comparisons for the main effect of Category revealed that the β-values for the human category were significantly higher compared to the β-values of all the other three categories (p≤0.005 for all comparisons).

For the late blind (LB)/SC comparisons, we run two separate ANOVA, one in the occipital ROI and one in the temporal ROI.

In the occipital ROI (from the univariate contrast LB > SC), we observed a significant main effect of Group ($F_{(1,33)}$=16.88; p=0.0002) while the main effect of Category ($F_{(3,99)}$=0.26; p=0.85) and the interaction Group*Category ($F_{(3,99)}$=0.82; p=0.48) were both not significant.

In the temporal ROI (from the univariate contrast SC > LB), we observed a significant main effect of Group ($F_{(1,33)}$=8.85; p=0.005) and a significant main effect of Category ($F_{(3,99)}$=23.93; p<0.001), while the interaction Group*Category ($F_{(3,99)}$=1.25; p=0.3) was not significant. The post hoc comparisons for the main effect of Category revealed that the β-values for the human category were significantly higher compared to the β-values of all the other three categories (p<0.001 for all comparisons).

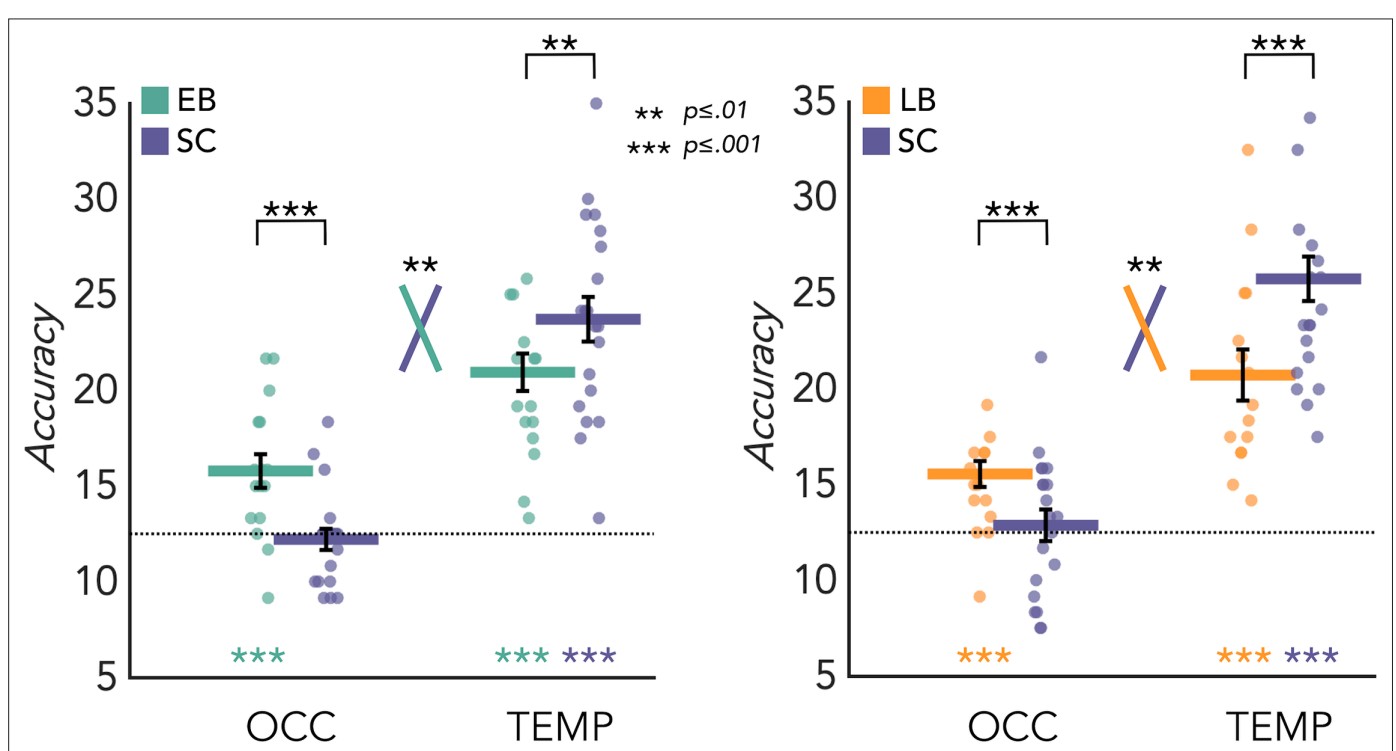

**Figure 3.** Multivoxel pattern (MVP) classification results in the regions of interest (ROIs). Eight-way decoding results from the early blind/sighted control (EB/SC) groups (left) and late blind (LB)/SC groups (right). Black bars show standard error (sample size: Early Blind=16; Late Blind=15; Sighted Controls=20).

The online version of this article includes the following figure supplement(s) for figure 3:

**Figure supplement 1.** Multivoxel pattern (MVP) classification results at the searchlight level.

**Figure supplement 2.** Multivoxel pattern analysis (MVPA) results for the late blind/sighted control (LB/SC) (age matched).

Note that in this analysis the groups' difference was expected, since the ROIs have been selected based on that and we will not further interpret this main effect of Group.

Interestingly, we did not find any significant interaction Group*Category, suggesting that the univariate data cannot point out the role of one specific category in driving the group differences between blind and sighted.

Does this mean that all the categories are equally represented in our ROIs in sighted and in blind groups? To better address this point and to look at the categorical representation of the different sounds at a finer-grained scale in both sighted and blind individuals, we run a further set of multivariate fMRI analyses.

## MVP eight-way classification

MVPA results for the EB/SC groups are represented in *Figure 3A* (left panel). In the SC group the mean decoding accuracy (DA) of the eight categories is significantly different from chance level (12.5%) in the temporal (DA = 23.71%; p<0.001) but not in the occipital (DA = 12.21%; p=0.67) ROIs. In the EB group the mean decoding accuracy is significant in both temporal (DA = 20.94%; p<0.001) and occipital cortex (DA = 15.78%; p<0.001). Importantly, a permutation test also revealed a significant difference between groups in both regions. In the occipital cortex the decoding accuracy value is significantly higher in EB than the SC (p<0.001, Cohen's d=1.25), while in the temporal ROI the accuracy value is significantly higher in SC than EB (p=0.01, Cohen's d=0.79). Importantly, the adjusted rank transform test (ART) 2 Groups × 2 ROIs revealed a significant group by region interaction ($F_{(1,34)}$=11.05; p=0.002).

MVPA results for the LB/SC groups are represented in *Figure 3A* (right panel). In the SC group the decoding accuracy is significant in the temporal (DA = 25.75%; p<0.001) but not in the occipital (DA = 12.87%; p=0.31) ROI. In the LB group the decoding accuracy is significant in both occipital (DA = 15.56%; p<0.001) and temporal (DA = 20.75%; p<0.001) regions.

A permutation test also revealed a significant difference between groups in both regions. In the occipital cortex the decoding accuracy value is significantly higher in LB than the SC (p<0.001, Cohen's d=0.73), while in the temporal ROI the accuracy value is significantly higher in SC than LB (p<0.001, Cohen's d=0.96). Importantly, the ART 2 Groups × 2 ROIs revealed a significant group by region interaction ($F_{(1,33)}$=7.154; p=0.01). We obtained similar results also when comparing the 15 late

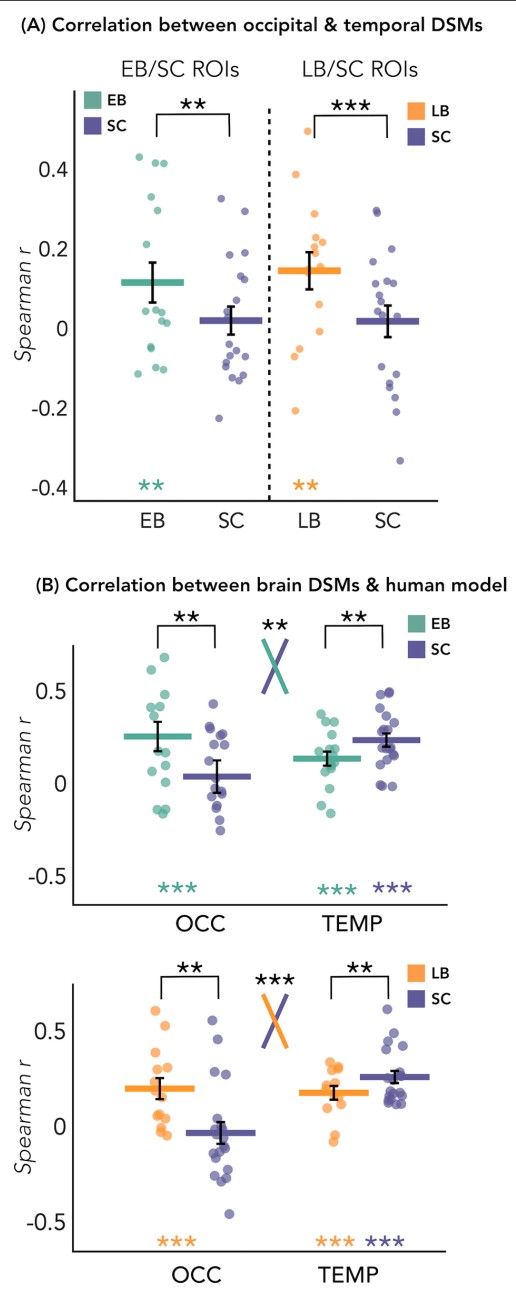

**Figure 4.** Dissimilarity matrices (DSMs) correlations. (**A**) Spearman's correlation between occipital and temporal DSMs. (**B**) Spearman's correlation between brain DSMs (occipital and temporal) and the human model. Black bars show standard error (sample size: Early Blind=16; Late Blind=15; Sighted Controls=20).

The online version of this article includes the following figure supplement(s) for figure 4:

**Figure supplement 1.** Correlation of brain dissimilarity matrices (DSMs) with behavioral models.

blind individuals with a subgroup of 15 age-matched sighted controls (*Figure 3—figure supplement 2*).

In addition, we also report the same analysis performed with a searchlight approach (*Figure 3—figure supplement 1*) for each group vs. baseline (p<0.05 FWE) and for each groups' comparison (p<0.005 unc) including the direct comparison of EB and LB. The results from this whole brain analysis support our ROI results. Indeed, we generally observed an enhanced decoding accuracy in the fronto-temporal areas in SC compared to blind groups (both EB and LB) and a decreased decoding accuracy in the occipito-parietal regions in SC compared to blind groups (both EB and LB).

## Representational similarity analysis

### RSA – correlation between the representational structure of occipital and temporal ROIs

The results of this analysis are represented in *Figure 4A*. We looked at whether the representation of the eight sound categories shares any similarity between the occipital and the temporal parcels within each blind and sighted subject, with particular interest at group differences. The permutation test revealed a significant correlation between the representational structure of occipital ROI and the representational structure of the temporal region only in blind groups (EB: r=0.12, p<0.01; LB: r=0.14, p<0.01), but not in SC group (r=0.02 in between both ROIs). When we look at the differences of correlations values between groups, we found a significant difference between the EB and the SC groups (p<0.01, FDR corrected), highlighting an increased similarity between the occipital and the temporal dissimilarity matrices (DSMs) in the EB when compared to the SC group (*Figure 4A*). The difference between the LB and the SC (*Figure 4A*) was also significant (p<0.001, FDR corrected), showing an increased similarity between the occipital and the temporal DSMs in the LB when compared to the SC group (*Figure 4A*).

### Comparison between brain DSMs and different representational models based on our stimuli space

Is there a specific feature that makes the structure of the occipital DSMs of blind closer to their temporal ROI DSMs?

Based on which dimensions (high or low level) are the sounds represented in the temporal and in the occipital parcels in our groups? The RSA comparisons with representational models, based either on low-level acoustic properties of the sounds or on high-level representations, can give us some important information about which representational structure could drive the observed decoding and correlation results.

The correlations' results with representational models are represented in *Figure 5C and D*.

In *Figure 5D* we reported the ranked correlation between the occipital DSMs in each group and each of the seven representational models. The human model showed the highest correlation with the DSM of the occipital ROIs in the blind groups (EB: r=0.20, p=0.0012; LB: r=0.16, p<0.014). In the SC group, none of the models shows a significant correlation with the occipital DSM. The r values and the p-values for each model and group are reported in *Supplementary file 4*. See also *Figure 5A and B* to visualize the complete set of models and the correlation between them.

In *Figure 5C* we reported the ranked correlation between the temporal DSM in each group and each of the seven representational models. For the temporal ROIs, the human model was the winning model in each group (in the SC > EB temporal ROI, SC: r=0.24, p<0.00002; EB: r=0.14, p<0.001; in the SC > LB temporal ROI, SC: r=0.26, p<0.00002; LB: r=0.18, p<0.001), explaining the functional profile of the temporal regions more than all other models with the exception of the behavioral model (see *Figure 5C*). In each group, the amount of correlation between the behavioral model and every temporal DSMs was quantitatively, but not significantly, lower compared to the human model (for the behavioral model in the SC > EB temporal ROI, SC: r=0.18, p<0.001; EB: r=0.13, p<0.001; in the SC > LB temporal ROI, SC: r=0.22, p<0.001; LB: r=0.16, p<0.001).

The r values and the p-values for each model and group are reported in *Supplementary file 5*.

Since the human model is the only one that significantly correlates with the occipital DSM in blind groups and that explains most of the variance of our data in the temporal ROI of each group, we ran further analyses for this model. That is, we directly investigated whether there was a statistical difference between groups in the correlation with the human model, both in occipital and in temporal ROIs.

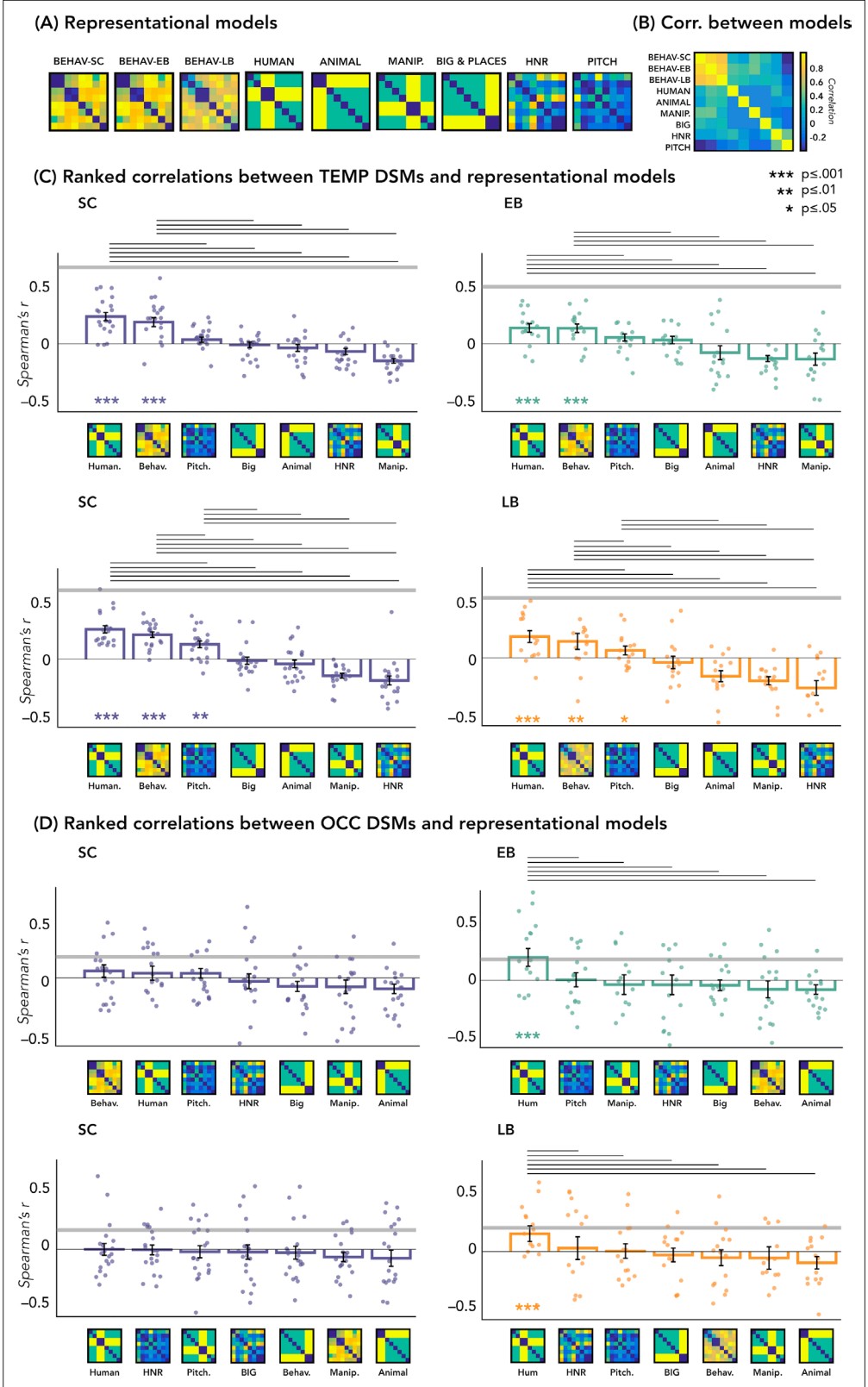

**Figure 5.** Representational similarity analysis (RSA) – correlations with representational models. (**A**) Representation of the seven models. (**B**) Matrix including the linear correlations between each pair of models. Yellow indicates high correlations, blue indicates low correlation. (**C**) Correlations between temporal dissimilarity matrix (DSM) of each group and the seven representational models. (**D**) Correlation between occipital DSM of each group

*Figure 5 continued on next page*

*Figure 5 continued*

and the seven representational models. Bars show mean Spearman's correlations across participants; error bars show standard error and each dot represents one participant (Sample size: Early Blind=16; Late Blind=15; Sighted Controls=20). Horizontal thick gray lines show the lower bound of the noise ceiling, which represents the reliability of the correlational patterns and provides an approximate bound of the observable correlations between representational models and neural data (*Bracci and Op de Beeck, 2016*; *Nili et al., 2014*). An asterisk below the bar indicates that correlations with that model were significantly higher than zero. Correlations with individual models are sorted from highest to lowest. Horizontal black lines above bars show significant differences between the correlations of the two end points (FDR corrected for multiple comparisons): we only reported the statistical difference between models showing a positive significant correlation and all other models.

The online version of this article includes the following figure supplement(s) for figure 5:

**Figure supplement 1.** Representational similarity analysis (RSA) results with the human model for the late blind/sighted control (LB/SC) (age matched).

RSA results with the human model for the EB/SC groups are represented in *Figure 4B* (top panel). In the occipital ROI, the permutation test revealed a significantly higher correlation in EB compared to the SC (p=0.019, Cohen's d=0.65). In the temporal ROI, the permutation test revealed a significantly lower correlation in EB compared to the SC (p=0.013, Cohen's d=0.76). Finally, ART analysis 2 Groups × 2 ROIs did reveal a significant effect of interaction group by region (p=0.007).

RSA results with the human model for the LB/SC groups are represented in *Figure 4B* (bottom panel). In the occipital ROI, the permutation test revealed a significantly higher correlation in LB compared to the SC (p<0.01, Cohen's d=0.72), while in the temporal ROI there was a significantly lower correlation in LB compared to the SC (p=0.012, Cohen's d=0.65). The ART analysis 2 Groups × 2 ROIs revealed a significant interaction between groups and regions (p<0.001).

To be sure that the behavioral model was not showing a similar trend as the human model, we perform an additional statistical analysis also for this model. With this supplemental analysis, we investigated whether there was a statistical difference between groups in the correlation with the behavioral model (see *Figure 4—figure supplement 1*). This analysis did not reveal any significant difference between groups nor an interaction Group*Region.

## RSA – additional whole brain searchlight analyses

We performed these analyses to show empirically that our stimuli are at least partially independent from the representation of low-level auditory properties in the temporal cortex and that they are suitable for investigating categorical auditory representation.

First, we used RSA with partial correlation to look whether we could segregate the representation of categorical vs. low-level auditory properties of our stimuli. In *Figure 6* it is shown how we could segregate in every group the portion of the temporal cortex representing the pitch regressing out human coding (on the right Heschl gyrus), a specific portion of right STG coding for harmonicity-to-noise ratio (HNR) in the three groups also regressing out human coding and finally the representation of the human model regressing out both pitch and HNR (bilateral superior lateral temporal region).

Second, we also run a split-half analysis to show that our auditory stimuli produce a stable pattern of activity in the temporal cortex (see *Figure 6—figure supplement 1*). Our data show that in all the groups (i.e. SC, EB, and LB) a big portion of the temporal cortex (including the superior temporal gyrus [STG], part of the middle temporal gyrus, and the Heschl gyrus both in the left and in the right hemispheres) show a highly significant stability of the patterns, suggesting that these portions of the temporal cortex have a stable representation of the sounds we selected. Interestingly, we also observed that the split-half correlation is increased in the occipital cortex of both EB and LB groups compared to the SC and it is, concomitantly, decreased in the temporal cortex of both blind groups (EB and LB) when compared to sighted subjects. A result that goes in line with our main decoding results.

## Discussion

Our study provides a comprehensive exploration of how blindness at different age of acquisition induces large-scale reorganization of the representation of sound categories in the brain. More

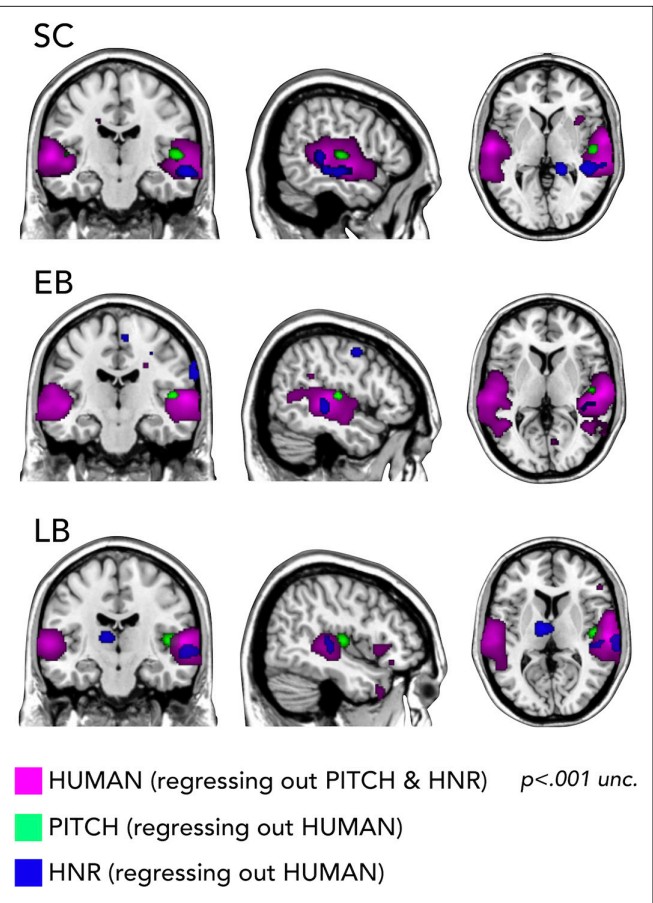

**Figure 6.** Representational similarity analysis (RSA) searchlight results with human, pitch, and harmonicity-to-noise ratio (HNR) models (including partial correlation).

The online version of this article includes the following figure supplement(s) for figure 6:

**Figure supplement 1.** Splithalf searchlight results.

precisely, compared to our previous paper on which we build on *Mattioni et al., 2020*, the present study sheds new lights on at least two fundamental issues: (1) How does the reorganization of occipital regions in blind people impact on the response profile of temporal regions typically coding for sounds, and (2) how does the age of blindness onset impact on those large-scale brain (re)organization.

First, we looked whether brain regions show a different level of activation for sounds in blind compared to sighted subjects. We observed that, indeed, a ventral occipital region in the right hemisphere showed enhanced activation for sounds in both EB and LB compared to sighted individuals while some portions of the temporal regions decrease their activation in EB and LB compared to the sighted group (see *Figure 2A*). However, univariate analyses did not allow to reveal neither if the different categories were discriminated one from each other in these regions, nor if one or multiple categories were more/less represented in those brain regions of blind vs. sighted (see *Figure 2B*).

To address these questions, we looked more in detail at the patterns of activity generated by the different sound categories in those reorganized occipital and temporal regions that emerged from the univariate group contrasts (EB vs. SC and LB vs. SC). Do these ROIs discriminate the different categories across sensory experiences (i.e. sighted, EB, LB)? If so, could we observe a difference between blind subjects and SC? Results from the eight-way MVP classification analysis show enhanced decoding accuracies in the occipital ROI of EB when compared to SC that was concomitant to reduced decoding accuracy in the temporal cortex of EB people (see *Figure 3*). Like what was observed in EB, LB also showed enhanced representation of sound categories in the occipital ROI compared to SC while the temporal cortex showed lower decoding in LB.

A dominant view in the field is that the reorganization of occipital regions is less functionally organized in late blindness than the one observed in early blindness (*Bedny et al., 2012*; *Collignon et al., 2013*; *Kanjlia et al., 2009*). Our results overturn this view by showing functionally specific coding of sound categories that is present in LB and increased compared to SC (see *Figure 3A* and Appendix 1). This has broad implications since it supports the idea that the increased representation of sound categories in the VOTC of EB and LB people could be an extension of the intrinsic multisensory categorical organization of the VOTC, that is therefore partially independent from vision in sighted as well (*Mattioni et al., 2020*; see also *Amedi et al., 2002*; *Ricciardi and Pietrini, 2011*; *Striem-Amit and Amedi, 2014*). Indeed, for such conceptual view to be true, late visual deprivation should maintain or even extend the non-visual coding that is already implemented in the occipital cortex of sighted people. Our data support this hypothesis, helping to fill this gap in the literature.

Importantly, the eight-way decoding analysis revealed differences in the categorical representation between sighted and blind subjects both in the temporal and in the occipital regions. We observed enhanced decoding accuracies in the occipital ROI of blind subjects when compared to controls and this enhanced representation of sound categories in the occipital cortex was concomitant to reduced decoding accuracy in the temporal cortex of blind people (see *Figure 3*).

Would this redistribution of computational load across temporal and occipital regions predict a representation of auditory categories in the occipital ROI that is more similar to the representation of the same auditory categories in the temporal regions in blind when compared to sighted? Our results suggest that this is indeed the case. In fact we show that within each blind subject, the correlation between the occipital and the temporal categorical representations is significantly higher compared to the SC (*Figure 4A*).

Which dimension of our stimuli does determine the response properties of the temporal and occipital regions to sounds? Is one category, among the others, driving these differences between groups? Moreover, is this alteration in the decoding explained by some higher-level representations (e.g. categorical, humanness) or by low-level acoustic features (pitch or HNR) specific to our sounds?

We addressed these questions by looking at which model, among several types based on different categorical (e.g. behavioral similarity judgment, human model, object model, etc.) and acoustic (e.g. harmonicity, pitch) dimensions, would better account for the representation of the auditory categories in the occipital and temporal regions in both sighted and blind subjects (*Figure 5*). In the temporal cortex, we found that in every group the best model was a 'human' model, in which human stimuli were considered similar between themselves and different from all other animate and inanimate stimuli (*Figure 5A and C*).

Interestingly, we also found that the human model, when compared to other models, showed the highest correlation with the representation of the auditory categories in the occipital ROI of both our blind groups but not of the SC (see *Figure 4B* and *Figure 5D*). This finding is well compatible with the spatial location of our ROIs which correspond to regions known to be involved in the processing of faces and voices, respectively (*Benetti et al., 2017*). Indeed, the two occipital ROIs are located within the fusiform gyrus/infero-temporal cortex, partially overlapping with the fusiform face area (*Julian et al., 2009*), while the two temporal ROIs are located within the STG, extending over the left and right temporal voice area (*Belin et al., 2004*).

We also show that it is only the representational structure of our 'human' model that is reduced in both LB and EB groups in temporal regions, but not the encoding of other categorical and low-level acoustic features which is similar across groups (*Figure 5C*). Interestingly, this result relates to the reversed group difference observed in the occipital regions where we find enhanced representation of the 'human' model in EB and LB people but no alteration in the representation of other categorical or acoustic features of the sounds (*Figure 5D*).

Importantly, we additionally show how our stimuli are well suited to address such question (*Figure 6*) by demonstrating, in all groups, that models of some acoustic features of our sounds (pitch, HNR) correlate more with the representational structure implemented in the Heschl gyrus (pitch) and in a specific portion of the right STS (HNR) while the 'human' model correlates more with the representational structure of an extended bilateral portion of STS (see also *Giordano et al., 2013*).

Why is that important? It shows for the first time that acoustic features of sound processing are not altered in the auditory region and not represented in the occipital cortex of EB and LB people. In contrast, the representation of a higher-level category 'Human/Voice' is reduced in temporal regions

and enhanced in occipital regions (*Figure 4B*). Showing that the same feature of our sounds is concomitantly reorganized in temporal and occipital cortices of both blind groups is filling in an important gap in the literature about how changes in the deprived and remaining senses relate to each other in case of early and late visual deprivation, providing a mechanistic view on the way plasticity expresses following blindness.

Could such a difference be driven by general factors like group differences in perceptual abilities, learning, and/or attention? If the different profile of the temporal cortex between blind and sighted individuals was driven by such general factors, one could have expected a difference not only for the representation of the human model but also for other representational models (e.g. sounds of objects or animals or low-level acoustic properties of the sounds). Our results show that this is not the case (see RSA correlations with representational models: *Figure 5*). Whether such specific alteration in the representation of voices relates to difference in the way blind and sighted people process voices (*Bull et al., 1983*; *Hölig et al., 2014*; *Klinge et al., 2009*) remains to be explored in future studies.

To summarize, we discovered that in both EB and LB the enhanced coding of sound categories in occipital regions is coupled with lower coding in the temporal regions compared to sighted people. The brain representation of the voice category is the one mostly altered in both EB and LB when compared to sighted people. This brain reorganization in blind people is mostly explained by the categorical aspects of the voice category and not by their acoustic features (pitch or harmonicity). An intriguing possibility raised by our results is that visual deprivation may actually trigger a redeployment mechanism that would reallocate part of the processing typically tagging the preserved senses (i.e. the temporal cortex for the auditory stimulation) to the occipital cortex deprived of its most salient visual input.

## Method
### Participants
Fifty-two participants involved in our auditory fMRI study: 17 EB (10 Female (F), 15 LB (4 F) and 20 SC 6 F).

EB participants were congenitally blind or lost their sight very early in life and all of them reported not having visual memories and never used vision functionally (*Supplementary file 1*). One EB participant was able to only perform two out of the five runs and was excluded from the analyses. The EB and SC were age (range 20–67 years, mean ± SD: 33.31±10.57 for EB subjects, range 23–63 years, mean ± SD: 35.1±8.83 for SC subjects; t(34)=–0.55, p=0.58) and gender ($X^2$ (1,36)=2.6; p=0.11) matched.

LB participants acquired blindness after functional visual experience (age of acquisition ranging 6–45 years old and number of years of deprivation ranging 5–43 years). All of them reported having visual memories and having used vision functionally (*Supplementary file 1*). The LB and SC were gender ($X^2$ (1,35)=0.03; p=0.87) matched but not age matched (range 25–68 years, mean ± SD: 44.4±11.56 for LB subjects, range 30–63 years, range 23–63 years, mean ± SD: 35.1±8.83 for SC subjects; t(33)=2.70, p=0.01). For this reason, in every parametric test in which we statistically compared the groups we included the age as regressor of non-interest. For the permutation test, we report in the main paper the results including all subjects but in the supplemental material we included the results or the same test including only a subset of 15 sighted subjects age and gender matched with the LB group (*Figure 3—figure supplement 1* and *Figure 5—figure supplement 1*) We did not find any relevant change in the results when the sighted were all included or when we included only a subset of them.

All the EB and 17 of the SC subjects were the same participants included in *Mattioni et al., 2020*, and in the current study we are re-analyzing these data used in our previous work. Importantly, the LB group and the three additional SC subjects were acquired in the same MRI scanner and in the same time period (July 2015–April 2016).

All participants were blindfolded during the task. Participants received a monetary compensation for their participation. The ethical committee of the University of Trento approved this study (protocol 2014-007) and participants gave their informed consent before participation.

## Materials and methods

Since this paper is submitted as a Research Advances format, it represents a substantial development that directly build upon a Research Article published previously by eLife (*Mattioni et al., 2020*). As for the journal recommendation, no extensive description of material and methods will appear when directly overlapping with our previous publication.

### Stimuli

A preliminary experiment was carried out to select the auditory stimuli. The detailed procedure is described in *Mattioni et al., 2020*.

The final acoustic stimulus set included 24 sounds from eight different categories (human vocalization, human non-vocalization, birds, mammals, tools, graspable objects, environmental scenes, big mechanical objects) that could be reduced to four superordinate categories (human, animals, manipulable objects, big objects/places) (see *Figure 1* and *Supplementary file 2*).

### Procedure

Before entering the scanner, each participant was familiarized with the stimuli to ensure perfect recognition. In the fMRI experiment each trial consisted of the same stimulus repeated twice. Rarely (8% of the occurrences), a trial was made up of two different consecutive stimuli (catch trials). Only in this case, participants were asked to press a key with the right index finger if the second stimulus belonged to the living category and with their right middle finger if the second stimulus belonged to the non-living category. This procedure ensured that the participants attended and processed the stimuli. Each pair of stimuli lasted 4 s (2 s per stimulus) and the inter-stimulus interval between one pair and the next was 2 s long for a total of 6 s for each trial. Within the fMRI session, participants underwent five runs. Each run contained three repetitions of each of the 24 stimuli, eight catch trials and two 20-s-long periods (one in the middle and another at the end of the run). The total duration of each run was 8 min and 40 s. The presentation of trials was pseudo-randomized: two stimuli from the same category (i.e. animals, humans, manipulable objects, non-manipulable objects) were never presented in subsequent trials. The stimuli delivery was controlled using MATLAB R2016b (https://www.mathworks.com) Psychophysics toolbox (http://psychtoolbox.org).

### fMRI data acquisition and analyses

#### fMRI data acquisition and pre-processing

We acquired our data on a 4T Bruker Biospin MedSpec equipped with an eight-channel birdcage head coil. Functional images were acquired with a T2*-weighted gradient-recalled echo-planar imaging (EPI) sequence (TR, 2000 ms; TE, 28 ms; flip angle, 73°; resolution, 3×3 mm; 30 transverses slices in interleaved ascending order; 3 mm slice thickness; field of view (FoV) 192×192 mm$^2$). The four initial scans were discarded to allow for steady-state magnetization. Before each EPI run, we performed an additional scan to measure the point-spread function (PSF) of the acquired sequence, including fat saturation, which served for distortion correction that is expected with high-field imaging.

A structural T1-weighted 3D magnetization prepared rapid gradient echo sequence was also acquired for each subject (MP-RAGE; voxel size 1 × 1 × 1 mm$^3$; GRAPPA acquisition with an acceleration factor of 2; TR 2700 ms; TE 4.18 ms; TI (inversion time) 1020 ms; FoV 256; 176 slices).

To correct for distortions in geometry and intensity in the EPI images, we applied distortion correction on the basis of the PSF data acquired before the EPI scans (*Zeng and Constable, 2002*). Raw functional images were pre-processed and analyzed with SPM12 (Welcome Trust Centre for Neuroimaging London, UK; http://www.fil.ion.ucl.ac.uk/spm/software/spm/) implemented in MATLAB (MathWorks). Pre-processing included slice-timing correction using the middle slice as reference, the application of temporally high-pass filtered at 128 Hz, and motion correction.

To achieve maximal accuracy in the coregistration and normalization in a common volumetric space, we relied on the DARTEL (Diffeomorphic Anatomical Registration Through Exponentiated Lie Algebra; *Ashburner, 2007*) toolbox. DARTEL normalization takes the gray and white matter templates from each subject to create an averaged template based on our own sample that will be used for the normalization. The creation of a study-specific template using DARTEL was performed to reduce deformation errors that are more likely to arise when registering single subject images to an unusually shaped template (*Ashburner, 2007*). This is particularly relevant when comparing blind and sighted

subjects given that blindness is associated with significant changes in the structure of the brain itself, particularly within the occipital cortex (*Dormal et al., 2016*; *Jiang et al., 2009*; *Pan et al., 2007*; *Park et al., 2009*).

## General linear model

The pre-processed images for each participant were analyzed using a general linear model (GLM). For each of the five runs we included 32 regressors: 24 regressors of interest (each stimulus), 1 regressor of no-interest for the target stimuli to be detected, 6 head-motion regressors of no-interest, and 1 constant. From the GLM analysis we obtained a β-image for each stimulus (i.e. 24 sounds) in each run, for a total of 120 (24 × 5) β-maps.

## Regions of interest

We used univariate analyses to select our ROIs.

First, we contrasted all the sounds vs. the baseline in each group and then we looked at groups' comparisons to find the regions that were more active for sounds vs. baseline in 1. EB more than SC; 2. SC more than EB; 3. LB more than SC; 4. SC more than LB.

To foreshadow the results (*Figure 2A*) we found a region in the right VOTC, mostly in the infero-temporal cortex and in the fusiform gyrus (group peak coordinates in MNI space: 48 –60–14) more active in EB compared to SC and a region in the right STG (group peak coordinates in MNI space: 58 –18–10) more active in SC than in EB.

Similarly, we found a region in the left VOTC, mostly in the fusiform gyrus and in the infero-temporal cortex (group peak coordinates in MNI space: 44 –76–18) more active in LB than in SC and a region in the left STG (group peak coordinates in MNI space: –48–14 0) more active in SC than in LB. The two regions in the left VOTC were partially but not completely overlapping so we created two different ROIs. Therefore in total we created four different ROIs, two of them from the comparison of EB and SC: occipital EB-SC and temporal SC-EB and the other two from the comparison of LB and SC: occipital LB-SC and temporal SC-LB. In the further multivariate analyses we computed, we used the first two ROIs to compare the EB and SC groups and the last two ROIs to compare the LB and SC groups.

Importantly, to avoid any form of circularity, we applied a leave-one-subject-out approach: for each subject we run the just mentioned univariate contrasts excluding the subject himself/herself from the analysis (e.g. for the EB1 the occipital ROI is defined as the contrast [all EB but EB1>all SC]).

Since the univariate analyses highlighted only a small portion of VOTC (i.e. part of the fusiform gyrus and the infero-temporal cortex) in the contrasts EB > SC and LB > SC, we decided to run a topographical univariate functional preference analysis, as a supplemental analysis, to have a more comprehensive view on the reorganization of the VOTC following blindness and the impact of blindness's onset on such reorganization (see Appendix 1 for detailed description of this analysis).

## β's extraction

Is one among our four main categories (i.e. animal, human, manipulable objects, and big objects and places) driving the group differences of the univariate results? To address this point we extracted the β-values in each ROI and group for every main category. Then, for each ROI we entered the β-values in a repeated measures ANOVA 2(Groups)*4(Categories). Note that in this analysis the groups' difference is expected, since the ROIs have been selected based on that and we will not further interpret the main effect of Group. We run this analysis to see if there is a significant interaction Group*Category, which would highlight the role of one category among the others in explaining the groups' differences.

## MVP eight-way classification

MVP classification analysis was performed using the CoSMoMVPA (*Oosterhof et al., 2016*) toolbox, implemented in MATLAB R2016b (MathWorks). We tested the discriminability of patterns for the eight categories using a support vector machine analysis. We performed a leave-one-run-out cross-validation procedure using β-estimates from four runs in the training set, and the β-estimates from the remaining independent run to test the classifier, with iterations across all possible training and test sets. This procedure was implemented in our ROIs (defined with a leave-one-subject-out procedure): in each cross-validation fold, we first defined from the training data the 40 most discriminative voxels

according to our eight categories (*De Martino et al., 2008*; *Mitchell et al., 2004*) and then we ran the MVP classification on this subset of voxels in the test data using the parameters described above.

The number of selected voxels (i.e. n=40) is based on the number of voxels of the smaller ROI (i.e. temporal SC-EB n=42 voxels). In this way, we could select the same number of voxels in each ROI and group.

Statistical significance of the classification results within each group was assessed using a non-parametric technique by combining permutations and bootstrapping (*Stelzer et al., 2013*). For each subject, the labels of the different categories' conditions were permuted, and the same decoding analysis was performed. The previous step was repeated 100 times for each subject. A bootstrap procedure was applied to obtain a group-level null distribution that is representative of the whole group. From each subject's null distribution, one value was randomly chosen (with replacement) and averaged across all participants. This step was repeated 100,000 times resulting in a group-level null distribution of 100,000 values. The statistical significance of our MVP classification results was estimated by comparing the observed result to the group-level null distribution. This was done by calculating the proportion of observations in the null distribution that had a classification accuracy higher than the one obtained in the real test. To account for the multiple comparisons, all p-values were corrected using false discovery rate (FDR) (*Benjamini and Hochberg, 1995*).

The statistical difference between each group of blind (EB and LB) and the SC group was assessed using a permutation test. We built a null distribution for the difference of the accuracy values of the two groups by computing them after randomly shuffling the group labels. We repeated this step 10,000 times. The statistical significance was estimated by comparing the observed result (i.e. the real difference of the accuracy between the two groups) to the null distribution. This was done by calculating the proportion of observations in the null distribution that had a difference of classification accuracy higher than the one obtained in the real test. To account for the multiple comparisons, all p-values were corrected using FDR (*Benjamini and Hochberg, 1995*).

To analyze the interaction between groups and regions, we also performed a non-parametric test: the ART (*Leys and Schumann, 2010*). ART is an advisable alternative to a factorial ANOVA when the requirements of a normal distribution and of homogeneity of variances are not fulfilled (*Leys and Schumann, 2010*), which is often the case of multivariate fMRI data (*Stelzer et al., 2013*). Importantly, we used the adjusted version of the original rank transformation (RT) test (*Conover and Iman, 1981*). In fact, the classical RT method loses much of its robustness as soon as the main effects occur together with one or several interactions. To avoid this problem, in the adjusted version the scores are adjusted by deducting the main effects and then analyzing separately the interactions (*Leys and Schumann, 2010*).

We performed two separate ART tests, one for each blind group. The first ART with regions (occipital and temporal) as within-subject factor and with SC and EB groups as between-subjects factor. The second ART with regions (occipital and temporal) as within-subject factor and with SC and LB groups as between-subjects factor.

## RSA – brain DSM

We further investigated the functional profile of the ROIs using RSA. This analysis goes a step further compared to the decoding analysis revealing how each region represents the different stimuli categories and whether the results obtained in the decoding analyses are mostly driven by several categorical/high-level properties of the stimuli or by their low-level acoustic features such as pitch or harmonicity. RSA is based on the concept of DSM: a square matrix where the columns and rows correspond to the number of the conditions (8×8 in this experiment) and it is symmetrical about a diagonal of zeros. Each cell contains the dissimilarity index between two stimuli (*Kriegeskorte and Kievit, 2013*). This abstraction from the activity patterns themselves represents the main strength of RSA, allowing a direct comparison of the information carried by the representations in different brain regions, different groups, and even between brain and models (*Kriegeskorte and Mur, 2012*; *Kriegeskorte et al., 2008b*).

First, we computed the brain DSMs for each ROI and in each subject. We extracted the DSM (*Kriegeskorte et al., 2008a*) in each ROI, computing the dissimilarity between the spatial patterns of activity for each pair of conditions. To do so, we first extracted in each participant and in every ROI the stimulus-specific BOLD estimates from the contrast images (i.e. SPM T-maps) for all the eight

conditions separately. Then, we used Pearson's correlation to compute the distance between each pair of patterns. Since the DSMs are symmetrical matrices, for all the RSA we use the upper triangular DSM (excluding the diagonal) to avoid inflating correlation values.

## RSA – correlation between occipital and temporal ROIs in each subject and group

When the sounds of our eight categories are presented, brain regions create a representation of these sounds, considering some categories more similar and others more different. Would visual deprivation have an impact on the structure of representation for sound categories in the occipital and temporal regions? Our hypothesis was that the similarity between the representation of the eight sound categories between temporal and occipital regions was enhanced in blind individuals compared to their SC. To test this hypothesis, we compared the correlation between the DSMs of the occipital and temporal ROIs in each group.

In each individual, we computed the Spearman's correlation between the occipital and temporal DSMs. We then averaged the values across subjects from the same group to have a mean value per group (*Figure 4A*).

For statistical analysis, we followed the procedure suggested by *Kriegeskorte et al., 2008a*. For each group, the statistical difference from zero was determined using permutation test (10,000 iterations), building a null distribution for these correlation values by computing them after randomly shuffling the labels of the matrices. Similarly, the statistical difference between groups was assessed using permutation test (10,000 iterations) building a null distribution for these correlation values by computing them after randomly shuffling the group labels. The p-values are reported after FDR correction (*Benjamini and Hochberg, 1995*).

## RSA – comparison between brain DSMs and representational models based on our stimuli space

Based on which dimensions (high-level/categorical or low-level acoustic properties) are the eight sound categories represented in the temporal and in the occipital ROIs in our groups? To address this question, we compared the representation of the sound categories in the two ROIs in each group with different representational models based either on low-level acoustic properties of the sounds or on high-level representations. Which of these models would better describe the representation of the sound stimuli in each region and group? Would the winning model (i.e. the model eliciting the highest correlation) be the same in the occipital and in the temporal region in (EB and LB) blind and in sighted subjects?

First of all, we built several representational models (see *Figure 5A*) based on different categorical ways of clustering the stimuli or on specific acoustic features of the sounds (computed using Praat, https://praat.en.softonic.com/mac).

Five models are based on high-level properties of the stimuli (models from 1 to 5) and two models are based on low-level properties of the sounds (models from 6 to 7) for a total of seven representational models (see *Figure 5A and B* to visualize the complete set of models and the correlation between them):

1. Behavioral model: it is based on the subject's ratings of similarity, which were based on categorical features. We included one behavioral model for each group.
2. Human model: it is a combination of a model that assumes that the human categories cluster together and all other categories create a second cluster and a model that assumes that the human categories cluster together and all other categories are different from humans and between themselves (*Contini et al., 2020*; *Spriet et al., 2022*).
3. Animal model: it is a combination of a model that assumes that the animal categories cluster together and all other categories create a second cluster and a model that assumes that the animals categories cluster together and all other categories are different from humans and between themselves.
4. Manipulable model: it is a combination of a model that assumes that the manipulable categories cluster together and all other categories create a second cluster and a model that assumes that the manipulable categories cluster together and all other categories are different from humans and between themselves.

5. Big and place model: it is a combination of a model that assumes that the big and place model categories cluster together and all other categories create a second cluster and a model that assumes that the big and place model categories cluster together and all other categories are different from humans and between themselves.
6. HNR model: the HNR represents the degree of acoustic periodicity of a sound.
7. Pitch model: the pitch, calculated with the autocorrelation method (see *Mattioni et al., 2020*), represents the measure of temporal regularity of the sound and corresponds to the perceived frequency content of the stimulus.

Then, we computed the Spearman's correlation between each model and the DSM of each subject from the occipital and from the temporal ROIs, using a GLM approach. For each region separately, we finally averaged the correlation values of all subjects from the same group (*Figure 4C and D*).

Statistical significance of the correlation results within each group was assessed using a non-parametric technique by combining permutations and bootstrapping (*Stelzer et al., 2013*), as we did for the decoding analyses (for further details about this statistical analysis, see the section above: MVP eight-way classification).

To account for the multiple comparisons, all p-values were corrected using FDR correction across the seven comparisons for each ROI (*Benjamini and Hochberg, 1995*).

To partially foreshadow the results, this analysis revealed that the human model is the winner model in the temporal ROI of each group and in the occipital ROI of blind groups. Therefore, only for the human model we performed statistical analyses to look at the comparison between groups (EB vs. SC and LB vs. SC) in both temporal and occipital ROIs (*Figure 4B*).

The statistical difference between each group of blind (EB and LB) and the SC group was assessed using a permutation test. We built a null distribution for the difference of the correlation values of the two groups by computing them after randomly shuffling the group labels. We repeated this step 10,000 times. The statistical significance was estimated by comparing the observed result (i.e. the real difference of the correlations between the two groups) to the null distribution. This was done by calculating the proportion of observations in the null distribution that had a difference of correlation higher than the one obtained in the real test.

Similar to the MVP eight-way classification analysis, we performed the non-parametric ART to analyze the interaction between groups and regions (*Leys and Schumann, 2010*).

## RSA – additional whole brain searchlight analyses

We run some further analyses to show empirically that our stimuli are suitable for investigating categorical auditory representation and that this categorical representation is at least partially independent from the representation of low-level auditory properties in the temporal cortex.

First, we used RSA with partial correlation to look whether we could segregate the representation of human model vs. low-level auditory properties (pitch and HNR) of our stimuli. To do so, we compute the correlation between the brain representation of our stimuli with either a human, a pitch, or an HNR model regressing out the partial correlation shared between these models (*Figure 6*).

Second, we run a split-half analysis combined with a searchlight approach to show that our auditory stimuli produce a stable pattern of activity in the temporal cortex (see *Figure 6—figure supplement 1*). We split the data in two halves, and we computed in each sphere of the brain a value of stability of the pattern of activity produced by the sounds. To do so we created for each voxel a matrix including for each stimulus the correlation between the patter of activity that such a stimulus produced with the pattern of activity produced by all other stimuli. In our case it is a 24*24 matrix, since we have 24 sounds in total. Then, we computed the average of the on-diagonal values minus the average of the off-diagonal values and we use the obtained value as the 'stability value'.

## Acknowledgements

This work was supported by a European Research Council starting grant (MADVIS grant #337573) attributed to OC, the Belgian Excellence of Science program (EOS Project No. 30991544) attributed to OC and the Flag-ERA HBP PINT-MULTI (R.8008.19) attributed to OC and a mandate d'impulsion scientifique (MIS-FNRS) attributed to OC. MR is a research fellow and OC a research associate at the National Fund for Scientific Research of Belgium (FRS-FNRS). Computational resources have been provided by the supercomputing facilities of the Université catholique de Louvain (CISM/UCL) and the

Consortium des Équipes de Calcul Intensif en Fédération Wallonie Bruxelles (CÉCI) funded by the Fond de la Recherche Scientifique de Belgique (FRS-FNRS) under convention 2.5020.11 and by the Walloon Region. We are thankful to our blind participants and to the Unioni Ciechi of Trento, Mantova, Genova, Savona, Cuneo, Torino, Trieste, and Milano and the blind Institute of Milano for helping with the recruitment. We are also grateful to Jorge Jovicich for technical assistance in developing fMRI acquisition sequences and to Roberto Bottini for the help with blind participants' recruitment.

## Additional information

### Funding

| Funder | Grant reference number | Author |
| --- | --- | --- |
| European Research Council | 337573 | Olivier Collignon |
| Belgian Excellence of Science | 30991544 | Olivier Collignon |
| Flag-ERA HBP PINT-MULTI | 8008.19 | Olivier Collignon |
| Mandate d'impulsion scientifique MIS - FNRS | | Olivier Collignon |
| National Fund for Scientific Research of Belgium | | Olivier Collignon |
| Fond National de la Recherche Scientifique de Belgique (FRS-FNRS). | | Ceren Battal |

The funders had no role in study design, data collection and interpretation, or the decision to submit the work for publication.

### Author contributions

Stefania Mattioni, Conceptualization, Data curation, Formal analysis, Investigation, Visualization, Methodology, Writing – original draft, Writing – review and editing; Mohamed Rezk, Ceren Battal, Jyothirmayi Vadlamudi, Data curation, Writing – review and editing; Olivier Collignon, Conceptualization, Resources, Data curation, Supervision, Funding acquisition, Validation, Investigation, Methodology, Writing – original draft, Writing – review and editing

### Author ORCIDs

Stefania Mattioni http://orcid.org/0000-0001-8279-6118
Mohamed Rezk http://orcid.org/0000-0002-1866-8645
Ceren Battal http://orcid.org/0000-0002-9844-7630
Olivier Collignon http://orcid.org/0000-0003-1882-3550

### Ethics

Human subjects: The ethical committee of the University of Trento approved this study (protocol 2014-007) and participants gave their informed consent before participation.

### Decision letter and Author response

Decision letter https://doi.org/10.7554/eLife.79370.sa1
Author response https://doi.org/10.7554/eLife.79370.sa2

## Additional files

### Supplementary files

- Supplementary file 1. Characteristics of early and late blind participants.
- Supplementary file 2. Categories and stimuli.
- Supplementary file 3. Groups' mean β-values for each region of interest (ROI) and for every category.

• Supplementary file 4. R- and p-values (false discovery rate [FDR] corrected for seven comparisons) from representational similarity analysis (RSA) correlation between the occipital dissimilarity matrices (DSMs) and representational models.

• Supplementary file 5. R- and p-values (false discovery rate [FDR] corrected for seven comparisons) from representational similarity analysis (RSA) correlation between temporal dissimilarity matrices (DSMs) and representational models.

• Transparent reporting form

### Data availability

Raw data are not provided as personal consent was not obtained in each participant for their data to be made publicly available. This is especially sensitive as the raw data contain anatomical MRI scans of the participant and therefore full anonymity cannot by default be guaranteed even if we deface those images. Due to those restrictions imposed on data sharing in our ethical approval (anonymity should be fully guaranteed), defaced raw MRI data can only be shared upon request to the corresponding author Olivier Collignon (olivier.collignon@uclouvain.be). Olivier Collignon will evaluate if the request come from an academic team with a specific scientific question in mind. If those criteria are met, the data will be shared. These data cannot be provided for commercial research. Processed data (statistical maps), numerical data and Matlab scripts have been made open on OSF database reachable using this link https://doi.org/10.17605/OSF.IO/FEQA6.

The following dataset was generated:

| Author(s) | Year | Dataset title | Dataset URL | Database and Identifier |
|-----------|------|---------------|-------------|-------------------------|
| Mattioni S | 2022 | Impact of blindness onset on the representation of sound categories in occipital and temporal cortices | https://doi.org/10.17605/OSF.IO/FEQA6 | Open Science Framework, 10.17605/OSF.IO/FEQA6 |

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

## Appendix 1

## Topographical univariate functional preference maps
### Methods
Since the univariate analyses highlighted only a small portion of VOTC (i.e. part of the fusiform gyrus and the infero-temporal cortex), we decided to run this supplemental analysis to have a more comprehensive view on the impact of blindness' onset on the reorganization of the VOTC.

In the topographical analysis, we also used additional data from a visual version of the experiment.

We created a visual version of the stimuli set. The images for the visual experiment were colored pictures collected from Internet and edited using GIMP (https://www.gimp.org). Images were placed on a gray 400×400 pixels background.

An additional group of 16 sighted participants (SCv) took part in this visual version of the experiment (see *Mattioni et al., 2020*, for further details).

We created a topographical functional preference map for VOTC ROI for each group. We also included the maps from the additional group of sighted that performed a visual version (SCv) of the same experiment. The VOTC ROI included the Fusiform, the Parahippocampal, and the Infero-Temporal cortices.

To create the topographical functional preference map (*Figure 2*), we extracted in each participant the β-value for each of our four main conditions (animals, humans, manipulable objects, and places) from each voxel inside each mask and we assigned to each voxel the condition producing the highest β-value (winner takes all). This analysis resulted in specific clusters of voxels that spatially distinguish themselves from their surround in terms of preference for a particular condition (*van den Hurk et al., 2017*; *Mattioni et al., 2020*).

Finally, to compare how similar are the topographical functional preference maps in the four groups we followed, for each pair of groups [(1) SCv-EB; (2) SCv-SC; (3) SCv-LB; (4) SC-EB; (5) SC-LB; (6) EB-LB] these steps: (1) We computed the Spearman's correlation between the topographical functional preference map of each subject from Group 1 with the averaged topographical functional preference map of Group 2 and we computed the mean of these values. (2) We computed the Spearman's correlation between the topographical functional preference map of each subject from Group 2 with the averaged functional preference map of Group 1 and we computed the mean of these values. (3) We averaged the two mean values obtained from Steps 1 and 2, to have one mean value for each group comparison. To test statistical differences, we used a permutation test (10,000 iterations). (4) We randomly permuted the conditions of the vector of each subject from Group 1 and of the mean vector of Group 2 and we computed the correlation (as in Step 1). (5) We randomly permuted the conditions of the vector of each subject from Group 2 and of the mean vector of Group 1 and we computed the correlation (as in Step 2). Importantly, we constrained the permutation performed in the Steps 4 and 5 to take into consideration the inherent smoothness/ spatial dependencies in the univariate fMRI data. In each subject, we individuated each cluster of voxels showing preference for the same category and we kept these clusters fixed in the permutation, assigning randomly a condition to each of these predefined clusters. In this way, the spatial structure of the topographical maps was kept identical to the original one, making very unlikely that a significant result could be explained by the voxels' spatial dependencies. We may however note that this null distribution is likely overly conservative since it assumes that size and position of clusters could be created only from task-independent spatial dependencies (either intrinsic to the acquisition or due to smoothing). We checked that each subject has at least seven clusters in his topographical map, which is the minimal number to reach the 10,000 combinations needed for the permutation given our four categories tested (possible combinations = $n\_categories^{n\_clusters}$; $4^7 = 16,384$). (6) We averaged the two mean values obtained from Steps 4 and 5. (7) We repeated these steps 10,000 times to obtain a distribution of correlations simulating the null hypothesis that the two vectors are unrelated (*Kriegeskorte et al., 2008a*). If the actual correlation falls within the top α×100% of the simulated null distribution of correlations, the null hypothesis of unrelated vectors can be rejected with a false-positives rate of α. The p-values are reported after FDR correction (for six comparisons).

To test the difference between the group pairs' correlations (we only test if in VOTC the correlation between the topographical maps of SCv and EB was different from the correlation of SCv and SC and if the correlation between SCv and LB was different from the correlation of SCv and SC) we used

a permutation test (10,000 iterations). (8) We computed the difference between the correlation of Pair 1 and Pair 2: mean correlation Pair 1 – mean correlation Pair 2. (9) We kept fixed the labels of the group common to the two pairs and we shuffled the labels of the subjects from the other two groups (e.g. if we are comparing SCv-EB vs. SCv-SC, we keep the SCv group fixed and we shuffle the labels of EB and SC). (10) After shuffling the groups' labels, we computed again the point 1-2-3 and 8. (11) We repeated this step 10,000 times to obtain a distribution of differences simulating the null hypothesis that there is no difference between the two pairs' correlations. If the actual difference falls within the top $\alpha \times 100\%$ of the simulated null distribution of difference, the null hypothesis of absence of difference can be rejected with a false-positives rate of $\alpha$.

## Results

*Appendix 1—figure 1* represents the topographical functional preference maps, which show the voxel-wise preferred stimulus condition based on a winner-takes-all approach (for the four main categories: animals, humans, small objects, and places) in VOTC.

We found that the topographical auditory preference maps of the EB (r=0.16, p=0.0001) and SC (r=0.09, p=0.0002) partially matched the visual map obtained in SC during vision. The correlation was also significant between the auditory maps in sighted and in EB (r=0.10, p=0.0001). These results replicate our previous results in *Mattioni et al., 2020*.

Importantly for the goal of the present study, we found similar results also in the LB group. The auditory topographic map of the LB subjects partially matched the visual topographic map obtained in SC during vision (r=0.17, p=0.0001) and correlated with the auditory topographic map observed in EB (r=0.11, p=0.0001).

The magnitude of the correlation between EB and SCv topographical category selective maps was significantly higher when compared to the correlation between SC in audition and SCv (p=0.003). Also in the case of late acquired blindness, the magnitude of correlation between LB and SCv was higher than the correlation between SC in audition and SCv (p=0.002).

As an additional information, we also computed a noise ceiling that could be useful to evaluate the correlation between the topographical maps of the different groups (*Bracci and Op de Beeck, 2016*; *Nili et al., 2014*). We calculated the Spearman's correlation of the topographical maps in the visual experiment between sighted subjects (r=0.42) and in the auditory experiment between sighted (r=0.10), EB (r=0.08), and LB subjects (r=0.14). These values represent the reliability of the correlational patterns and provide an approximate noise ceiling for the observable correlations between the topographical maps. As expected (since we are looking at the categorical preference in VOTC), the highest correlation is the one within the sighted subjects in the visual modality. It is not surprising that this reliability value is much lower in the three groups for the auditory modality. Indeed, the representation of sounds in VOTC was expected to be more variable than the representations of visual stimuli. This additional information is helpful in the interpretation of the correlation between groups. Even if they are modest, they, indeed, explain most of the variance under these noise ceilings.

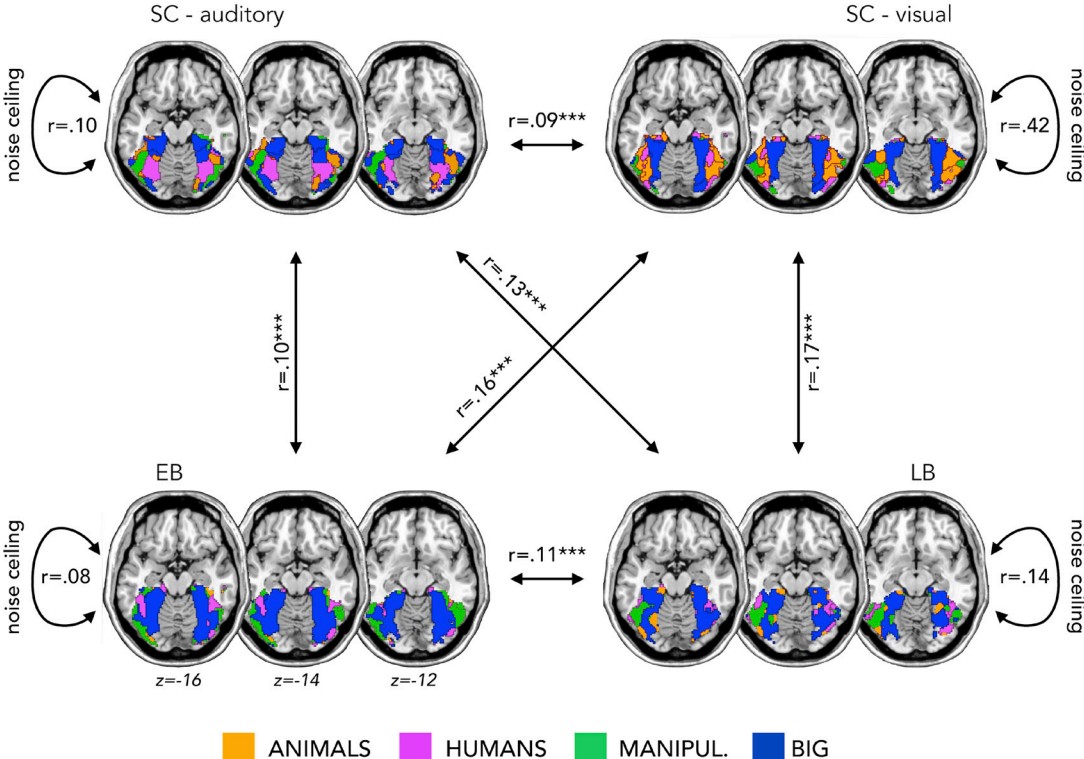

**Appendix 1—figure 1.** Topographical functional preference maps. Averaged 'winner take all' topographical functional preference maps for our four main categories (animals, humans, manipulable, big non-manipulable) in the auditory modality in the sighted controls (SC, top left), early blind (EB, bottom left), and late blind (LB, bottom right). In the top right we also reported the map from an additional group of sighted that performed the visual version of the experiment. These maps visualize the functional topography of ventral occipito-temporal cortex (VOTC) to the main four categories in each group. These group maps are created for visualization purpose only since statistics are run from single subject maps (see Materials and methods). To obtain those group maps, we first averaged the β-values among participants of the same group in each voxel inside the VOTC for each of our four main conditions (animals, humans, manipulable objects, and places) separately and we then assigned to each voxel the condition producing the highest β-value. For each group we also computed a noise ceiling value, computing the correlation of the map between subjects from the same group.

