## [Editor Report]

The study interrogates the representational structure of sound categories in the temporal cortex of early- and late-onset blind people. This adds two novel dimensions to the author's previous focused on auditory categorical representation in the visual cortex of people with early blindness onset, and as such will be of interest to researchers studying brain reorganisation across life. The strength of the study is in its methodology, which provides compelling and robust evidence to support the study's main conclusions.

---

## [Decision Letter]

**Decision letter after peer review:**

[Editors’ note: the authors submitted for reconsideration following the decision after peer review. What follows is the decision letter after the first round of review.]

Thank you for submitting the paper "Impact of blindness onset on the representation of sound categories in occipital and temporal cortices" for consideration by *eLife*. Your article has been reviewed by 3 peer reviewers, including Tamar R Makin as the Reviewing Editor and Reviewer #1, and the evaluation has been overseen by a Senior Editor.

Comments to the Authors:

We are sorry to say that, after much consultation with the reviewers, we have decided that this work will not be considered further for publication by *eLife*. This decision was reached because ultimately, we felt additional analysis will be required to address some of the key concerns raised, which might not produce sufficiently compelling results. However, should you feel that you are able to conclusively address the main comments made by the reviewers, and in particular the comments I highlight below, I will be happy to consider a resubmission of the manuscript.

As our decision was based on our panel discussion, for the sake of transparency I wanted to highlight the key issues that have been raised:

1. The ROI approach is too broad, and we want to understand where the information is coming from. This is particularly crucial for the temporal cortex ROI, which encompasses highly heterogenous sub-regions. As you will see below, there was a consensus that the paper contains too many overlapping analyses. With regards to the need for spatial clarity (e.g. searchlight), we agreed in our discussion that this analysis should focus on 2 key results:

(i) distances/decoding group differences – where in temporal cortex do they see reduced information content?

(ii) group differences in correlation with the visual category RDM (or the categories model) – where in the temporal cortex do they find greater sharing with visual cortex?

2. The paper includes too many different analyses which dilute the main findings and it has been our shared view that the balance of main (versus supplementary) analyses should be reconsidered. As you will see, there was some inconsistency across reviewers over which analyses you should focused on, and I believe that ultimately this should be the authors' call. But more careful editing of the Results section is necessary.

3. The results from the LB group are inconclusive and should be interpreted with more caution.

4. Perhaps our biggest issue was the use of stimuli based on visual functional organisation to study auditory representational structure, and compare this categories model to other models which are unlikely to be relevant to auditory cortex (e.g. manipulable objects). To clarify – we do know there are categorical responses in the auditory cortex, but as shown by other groups -they are anatomically constrained and in order to ensure they are indeed object-category responses, you need to better control for low-level auditory features. The fact that the ROIs are bilateral and huge (not only in size, but also in the diversity of selectivity of different subregions of the temporal cortex), combined with a model that does not represent the best response selectivity of the auditory cortex, can result in spurious findings. For example, let's say the best model for the right temporal cortex is pitch-based (very likely based on previous research), whereas for the posterior left is temporal or categorical. When you average them together these two may cancel out and push the results towards a different model or significant difference. We are not sure whether there's a way to solve this issue, but considering the key focus on auditory cortex we feel that there needs to be a better consideration of the native representational features before you can make any strong claims for crossmodal plasticity.

5. Your main observation for better decoding in the VOTC and worse decoding in the auditory cortex in blind participants is not novel, it has been reported in several papers, but the auditory cortex was usually treated merely as a control site (see Figure 4 in van den Hurk et al., 2017, PNAS; or Figure 4 in Vetter et al., Curr Bio, 2020). We agree that your findings extend and elucidate these original observations, and in my personal opinion the fact that this has been previously observed only strengthens your argument. But there needs to be greater transparency throughout the paper relating to what innovation the paper offers, and give credit to others when it is due. This also extends to the overlap with your previous *eLife* paper – none of the reviewers could understand if you are re-using your original dataset or replicating your previous findings.

*Reviewer #1 (Recommendations for the authors):*

The authors explore the information content (decoding accuracy) and representational structures of sound categories in the auditory cortex of individual with early- and late-onset blindness (EB/LB). They document a slight reduction in information content in auditory cortex in both early and late blinds relative to controls. They also find greater similarity in the representational structure of these sound categories with that of occipital cortex, which has already been shown in their previous work to be greater (more information) and also more similar to visual representation in the EB – this is extended here for the LB. The authors conclude that there's a redistribution of computational load which is not dependent on age of blindness onset. While this interpretation is speculative, it is quite interesting and the evidence, while modest, add novelty and context to the field.

A key advantage of the present study is the methods, including detailed accounts of the considerations underlying the various statistical tests which promotes both transparency and education of best statistical practices. However, considering the richness and density of the methods, this can be a double edged sward, and I find that a key disadvantage of the paper is that it is over-processed with multiple analyses, diluting the results of interest. This weakness should be easy to address. Another important advancement, relative to previous research, is the inclusion of the LB group. However, this too introduces a major issue as the effects of interest are quite small, and the study is likely to be underpowered to thoroughly address the impact of blindness onset. This could be addressed with more cautious interpretation of the findings.

1. Starting with the abstract and throughout the paper, the authors tend to make causal inferences from associations (e.g. reorganisation in occipital cortex impacts response profile in temporal cortex). But the observed findings in temporal cortex could be driven by different sources altogether (e.g. group differences in perceptual abilities, learning, attention, etc). These multiple drivers, as well as any behavioural differences in performance across groups, should be considered alongside the interpretation of a redistribution of computational load due to occipital cortex reorganisation. Similarly, non-significant results are sometimes taken as evidence for the null (e.g. nonsignificant interaction between EB and LB) and at other times as supporting the hypothesis (e.g. nonsignificant group difference between LB and controls in temporal cortex). Further evidence (e.g. Bayes Factors) should be provided in each case to better inform us about these findings.

2. Throughout the paper the authors split their control group to 2 (massively overlapping) control sub-groups. This seems unnecessary at best, and at its worst – introduces potential confounds. In particular, to compare between the two blind groups, the authors subtract the mean of each of the sub groups and divide this by the standard deviation of the relevant control sub group. But considering the groups are almost identical in their makeup, this normalisation is essentially taking into consideration the few control participants that were not included in both groups. I believe it makes better sense to add an age regressor and include all participants in the group comparisons. This way the regressor can account more accurately for potential impact of age on the dependent variables and in that respect – the broader the age range (e.g. in the controls, across the two blind groups) the more accurate the regressor. I would therefore encourage the authors to recalculate their group comparisons. Relatedly, I think that a direct comparison between the two blind groups would be valuable.

3. The "late" blind group ranged acquisition age of 6 to 45. This raises the question – should they be taken as one group? Losing sight in childhood is strikingly different from adulthood, as well as the progression of blindness etc. This adds a level of heterogeneity to this group that should be considered more carefully.

4. The ROIs that are used here are massive, and I'm left wondering where in the brain these effects occur. Further refinement of the spatial analysis (e.g. using a searchlight) would greatly improve our physiological understanding of the underlying mechanism.

I would recommend to tone down the use of the descriptive 'remarkable' (currently featured 5 times). (This is not a shampoo advertisement :).

Page 35 "This result suggests that the representation of auditory categories in the VOTC of early blind subjects is more similar to the representation of the same auditory categories in the temporal cortex of both sighted and early blind participants." – more similar than what?

*Reviewer #2 (Recommendations for the authors):*

The study used fMRI to investigate neural representations of sounds from various semantic categories (such as animals, humans, tools, big objects) in the brains of sighted, early blind, and late blind subjects. The authors report better classification of sounds into semantic categories in the early blind group than in the sighted group when the activation of the ventral occipitotemporal cortex (VOTC) is used in the analysis; an opposite effect was found for the superior temporal gyrus. Furthermore, the authors showed that the stimulus features represented in those two regions are more similar in the blind groups than in the sighted group. Based on these results, the authors propose that the absence of visual experience induces network-level reorganization in the brain – that in blind individuals the processing of "auditory categories" is partly delegated to the VOTC.

The research question investigated in the study is interesting and the adoption of multivariate analysis methods is timely, as the number of such studies in this research field is still relatively low. The univariate analysis and the classification analysis seem to partly replicate the findings that have been already reported, but the late blind group is added, which can be interesting in some research contexts. The final RSA analysis (Figure 8 and Figure 9) is very interesting and adds to our knowledge about mechanisms of brain reorganization in blindness.

The first issue that makes certain aspects of the study hard to evaluate is that the lack of important information about the study design in the Methods section. The authors write that the present study builds on research that is already published, and that they do not want to repeat information that is already there, which is fine. However, they should at least state if the fMRI experimental procedure was identical to the one that they used previously, and if the same or different subjects took part in the two studies. If the experimental procedure was the same, then it is unclear why the selection of new stimulus set was necessary (p. 7-8). If the procedure was different, then the reason for this change and the implications for the present study should be clearly described.

The second issue is that, at times, the manuscript is unclear in describing the theoretical background and the findings of the present study. Contrary to what is written in the manuscript, a few studies have already described differences in decoding accuracies in visual and auditory cortices across blind and sighted individuals (e.g. Dormal et al., 2016, Neuroimage; van den Hurk et al., 2017, PNAS; Vetter et al., 2020, Current Biology). Furthermore, the authors use terms "encoding of sounds in the VOTC" or "encoding of auditory categories in the VOTC" – yet, their RSA results seem to suggest the categorical representation in this region, both in blind and in the sighted subjects. The acoustic and categorical claim should be clearly differentiated (can the semantic category be visual or auditory? Do the authors mean that the VOTC in blind individuals capture acoustic properties that differentiate categories of sounds? Clearer definitions of terms such as "auditory category" or "functional relevance" would be helpful in understanding the authors' claims).

Thirdly, certain analytical choices could be better described. The authors write that they used broad VOTC and superior temporal ROIs. However, for the classification analysis, they decided to select only 250 voxels that were the most discriminative for a given comparison in the training data. Could the observed between-group difference be partly explained by the fact that this procedure resulted in the actual ROIs that were systematically different across groups, for the reasons not related to the imbalance across the sensory cortices? Or because the representation of certain stimuli in the blind group is expanded? (e.g., optimal decoding of presented sounds in the auditory cortices requires 300 voxels in this group due to the experience-dependent expansion of sound representation). Secondly, the interpretation of the univariate selectivity maps (should be "univariate functional preference maps") in their current form is difficult. The authors report correlation between maps obtained in the auditory modality in the blind group and in the visual modality in the sighted group, but the correlation strength is quite modest and there are clear qualitative differences between the maps in both groups (e.g., no clear preference for animal sounds and a different pattern of preference for human sounds in the blind groups, particularly in the early blind subjects). Inclusion of some sort of noise ceiling (e.g., constructed by correlating visual maps in two sighted groups) would be very helpful in evaluating this analysis.

Finally, what should the reader think about the findings in the late blind group? In the introduction the authors describe the two views of the brain reorganization following late blindness (p. 5; are they mutually exclusive?). However, it seems that the present findings can be well accommodated by either of the views. This aspect of the data should be better discussed. Furthermore, "functional relevance" and "more/less epiphenomenal" are quite unfortunate terms. Based on the current results, the authors cannot claim that the described neural representations in the VOTC are functionally relevant for subjects' behavior.

Despite the above-described issues, the reported findings, and particularly the results of the RSA analysis, are an interesting contribution to the debate on the mechanisms of neural plasticity in blindness. The study is so far the most convincing demonstration that blindness affects the representational content of not only the high-level visual cortex (the VOTC), but also the auditory cortex. However, certain aspects of the data analysis and of the claims that are being made can be clarified and improved.

Please see below for a couple of additional suggestions.

1) What is the noise ceiling in Figure 8? This information seems to be missing.

2) Page 33: the authors found that there is no correlation between subjects, within the one of the sighted groups, for the VOTC dissimilarity matrices. How to interpret this result? If the VOTC in each sighted subject represents completely different information in the present study, then how can we interpret other results for this region in this group?

3) The inclusion of the analyses reported in Figures 5-7 in the main text should be better justified – right now, it is unclear if they significantly contribute to the authors' main claims, which creates an impression that the manuscript is overloaded with analyses.

**Note from the editor: I've asked R2 for a clarification on their 2nd point in the review, and they added the following:

- My comment about the encoding of auditory categories was about clarification, and might be partly related to R3's comment. The STG is sensitive to acoustic features but, at some point, can categorize stimuli into more "semantic" categories, for example human voices. How the STG does this is not clear, but a reasonable (at least to me) assumption is that this "semantic" categorization is at least partly driven by acoustic features – for example, all human voices might have some common temporal and spectral properties and this commonality is captured by certain areas within the STG. Do they propose that, in the blind, the VOTC starts to categorize stimuli based on acoustic features and that this is why this region becomes more similar to the STG (seems to be suggested by "share of computational load" hypothesis and the RSA)? It is not clear to me from the manuscript.

*Reviewer #3 (Recommendations for the authors):*

The paper by Mattioni et al. 2021 studies the effect of blindness on the reorganization of sensory regions of the brain. The paper builds on Mattioni et al. 2020, where the authors used multivariate methods to show categorical representations for sounds in visual regions (ventral-occipito temporal cortex VOTC) in early blind individuals.

In the present study, the authors expand this research by addressing two main aims:

1) Characterize the effect of onset of blindness on the reorganization of the temporal (auditory) and ventro-occipital (visual) cortex. This is important for our understanding of how sensory experience modulates the organization of sensory cortices, and whether a very early onset of sensory absence has a different effect from a later onset. It provides insights into the importance of developmental sensitive periods, testing whether blindness only results in brain reorganization when occurs early in life, or whether it can cause reorganization at any point, suggesting that sensory experience constantly modulates the functional organization of sensory cortices.

2) Understand how sensory experience influences intramodal plasticity, that is, the organization of the sensory cortices that process sensory inputs from the preserved senses. In particular, this paper tries to go beyond simply stating whether responses in the preserved sensory cortex are different between blind and sighted, and aims to understand whether and how the representation of categorical information is modulated by blindness.

The paper is ambitious in its goals and design. The authors recruited three groups of participants, including two groups of blind individuals (early and late onset), which is commendable given the challenges of recruiting such populations.

The paper builds on the authors previous work, showing that the use of multivariate techniques, in particular representational similarity analysis (RSA), can provide unique insights about crossmodal and intramodal plasticity and the representation of information, addressing gaps in our current understanding of neural reorganization.

The main limitations are in the selection of a model and ROIs for evaluating intramodal plasticity and representations in the temporal cortex. The results and conclusions rely very strongly on their choice of an object category model of representation, which does not necessarily represent the selectivity of the temporal cortex or the main dimension of variability of the stimuli set.

Control of acoustic features in the stimuli is lacking. The first set of analyses reported in the paper, from Figures1-4, assume that the only or most important dimension in which the stimuli are different is their categories into different subsets of objects. Based on the models reported in Figure 8 and information in the methods, many low-level acoustic features vary across categories. These other acoustic features are very likely to drive the responses of the temporal cortex. The choice of the object categorical models seemed to be based on what we know about the VOTC, where these categories and the responses of the VOTC have been characterized in many studies of visual object perception. As the authors mention in the discussion, much less is known about object category organization in the auditory cortex. As such, having stimuli that vary in their spectrotemporal characteristics across categories, variations that are very likely to be represented in the auditory cortex, I am not convinced object category models are the best choice.

The authors chose very large ROIs to conduct their analyses. This is problematic, because it is forcing a single outcome (in terms of selectivity, model fitness, classification, etc) from regions that are likely to have different selectivity. This is particularly problematic for the temporal cortex, where the selectivity of anterior and posterior regions varies significantly. Furthermore, averaging right and left ROIs is also problematic for the analysis of the superior temporal cortex, where the right and left have different selectivity for temporal and spectral processing, respectively.

In their conclusions, the authors suggest that specialization for other senses in regions usually considered 'unisensory' is what allows crossmodal plasticity in cases of sensory deprivation. For example, cortical regions typically considered to be 'visual' areas also show some specificity for processing auditory information, and this specialization is the foundation of crossmodal plasticity effects. This is supported by a similarity in the representation of categories of sounds and images in the VOTC in sighted and blind individuals (both reported here and in Mattioni et al., 2020). However, it is difficult to conclude whether this is also the case in late blind, because even though there is a similar trend a trend, some of the differences do not reach statistical significance.

Furthermore, results from the temporal cortex do not show the same selectivity for auditory and visual stimuli. However, this could be due to the author's choice of a model that does not best represent the selectivity of such cortex, and alternative models should be tested to support this conclusion.

These are my main comments about improvements to the manuscript:

1) The authors conducted a remarkably challenging and ambitious research, with a lengthy and complex set of analyses. I suggest guiding the reader through the relevance of each of the analyses, and reconsider whether they all add to the conclusions, or whether some are redundant. I suggest highlighting from the beginning the most important and innovative results. In my view, those are the RSA results, examining the fitness of different models, and looking at correlations between groups and ROIs.

Some small things like adding titles or descriptions within figures (not only in the legends), spelling out some of the acronyms in the titles, will also help the reader in quickly understanding the aim of the analysis and the differences between figures.

2) I suggest to remodel the Results section. One could first address the question the authors mention on page 51 of their discussion: 'Which dimension of our stimuli may determine the response properties of the temporal ROI'. I suggest starting the Results section showing the analysis reported in figure 8 and then use the best models to conduct the rest of the analysis, including correlations between groups and ROIs, as well as the topographical selectivity maps (I understand this will not be possible for models that are not categorical).

3) The MVPA analysis is also based on an object category representation. It is not clear what this analysis adds to the RSA analysis, and again, it is assuming object category is an important dimension. I will recommend removing this analysis from the paper.

4) The authors used very large ROIs for their analysis, and the main issues of this have been explained in the public review. There are several things that can be done to improve this without acquiring more data:

1) analyze right and left ROIs separately (it will not solve all the problems, but it will be a significant improvement);

2) do a searchlight analysis;

3) use ROIs from the Destrieux Atlas (instead of the Desikan-Killiany), and either correct for multiple comparisons or focus on specific temporal regions, such as PT or posterior temporal regions.

You can always use one of your functional runs as a functional localizer, but of course, this will significantly reduce the amount of data in your analysis.

[Editors’ note: further revisions were suggested prior to acceptance, as described below.]

Thank you for resubmitting your work entitled "Impact of blindness onset on the representation of sound categories in occipital and temporal cortices" for further consideration by *eLife*. Your revised article has been reviewed by three peer reviewers, one of whom is a member of our Board of Reviewing Editors, and the evaluation has been overseen by Tamar Makin as the Senior Editor.

The manuscript has been improved but there are some remaining issues that need to be addressed, as outlined below:

Essential revisions:

As you will see below, the reviewers were generally happy with your revisions in response to the original reviews. However, two new issues have emerged that in particular require further substantial revisions.

1) Circular analysis. The reviewers have had an extensive discussion on whether or not your ROI selection criteria might have potentially biased the group results. To cut a long discussion short, we acknowledge that the multivariate analysis is done between categories within individual subjects, whiles the ROI is defined based on the univariate group differences. However, the independence of the classifier from the univariate results is not guaranteed, and we could come up with plausible scenarios under which your ROI selection criteria artificially inflates the classifier group differences. For this reason, we think that at the very least you will need to repeat the ROI definition using a leave-one-out approach (that is, while excluding the subject on which the classification analysis is carried out). We expect that this will not substantially affect your results, but will avoid the issue of circularity.

2) The new focus on the human model is not well motivated by your experimental design (or your previous study), and at worst could be taken as HARKing. We are more than happy to discuss with you any potential solutions to this issue, but a straightforward one will be to include the full analysis from the original manuscript.

*Reviewer #1:*

The authors have done a fantastic job thoroughly addressing all of my comments from the previous submission. The paper is much easier to read now, and for this reason, its novelty and impact shines even brighter.

Unfortunately, the revised version of the manuscript raises a few key methodological and conceptual issues concerning circularity that will need to be ironed out. I hope that those could be addressed with further revisions. Please note that I've restricted my comments to any changes that have been made to the original submission.

ROI selection

While I understand the conceptual motivation to focus on the areas where there are noted group differences, I'm puzzled by the methodological implementation. ROI selection was based on group differences in activity for the main stimulus. Clearly, greater activity will lead to greater decoding abilities (because there's less information/signal in the control group – see figure 2). So when the same data is used for both defining the ROI and running the decoding analysis, this seems entirely circular to me. To overcome this circularity, the analysis requires a leave on out/split half approach.

Encoding analysis

While I found the narrative of teasing apart high level versus low level contributions appealing, i could not understand why 'humanness' was chosen as the high level model. The study was clearly not designed to address this question, as the categories are not equally distributed between human and non human sounds. The study was designed based on categorical similarities/differences, as clearly indicated in the colour code of Figure 1, and this should be the competing model to the low-level ones. Alternatively, high-level representational structure is often derived by the experiential self report of participants.

Reading the results I realise the authors observed a greater univariate group difference in the human categories and this has likely drove the decision to use the human model in further analysis. But I'd again argue here for circularity (see above) and ask that this is addressed in the analysis.

Key hypothesis

In the introduction, the authors set up the main motivation of the current study: "Would the same categorical representation be the one that could be reorganized in the temporal cortex of these blind individuals? If true this would speak up for an interplay between the features that are reorganized in the temporal and occipital cortices of visually deprived people". Based on this interesting framework, the representational structure of sounds in OTC and TC is shared. But the key analysis – a group comparison of the correlation across the RDMs of the two brain areas is not shown to us.

Open question

The fact that the low-level auditory models did not capture significant variance in the temporal cortex (and in fact seemed to perform similarly, if not better in the visual cortex of sighted controls) calls for a more serious characterisation of the brain area under investigation in control participants.

Figure 3C seems to me circular to Figures 2 and 3B, and I suggest removing it.

In the 'stability analysis' for the searchlight analysis, I couldn't quite understand why not run the same decoding analysis used for the main analysis (Figure 3)? As a side note – do the areas identified here in OTC actually overlap spatially with the ROI used in the main analysis?

As a side point, is it of any relevance/interest that the temporal ROI is in a different hemisphere for each group?

I was missing a direct comparison between the two blind groups to really bring home the message that they are not different from one another. Here, of course, some care should be taken into demonstrating evidence to support the null hypothesis (e.g. BF).

*Reviewer #2:*

Thank you very much to the authors for their effort in reviewing their paper. It has improved significantly, and the aims and rationale for the different analyses are much clearer.

The paper relies strongly on the results obtained with the RSA and MVPA analysis, but I have concerns about the circularity in the definition of ROIs, which bring to question the reliability of the results. I disagree that the definition of the ROI is not circular. The authors define the ROIs on differences across groups, and then use these ROIs to show that differences in classification across groups. This is circular, as in both cases the authors are looking at group differences. For example, in the MVPA analysis, a difference in intensity will also result in categorical classification differences. This, combined with the fact that differences between groups in the searchlight analysis are not significant at corrected level, puts in doubt the claim about reduced classification in the temporal cortex in blind individuals.

In addition, it is not clear how the authors went from a variety of models in their original manuscript, to the three models displayed in Figure 4. It is difficult to believe that these models capture the full variability of their stimuli. Take for example the results of Figure 6, where the "Human" model is the one that captures the best activity across most of the STC. It is known that STC does not only code 'human' vs 'non-human', which highlights that there is information missing in the models used.

*Reviewer #3:*

The authors answered my concerns, thank you for the detailed responses. I have only one writing suggestion:

– "Studying the same participants" does not necessarily mean "re-analysing data used in our previous work". I would recommend clarifying in the paper that the data from CB and SC participants are the data that were also analysed in the previous paper (i.e., it is a reanalysis).

---

## [Author Response]

[Editors’ note: the authors resubmitted a revised version of the paper for consideration. What follows is the authors’ response to the first round of review.]

We are sorry to say that, after much consultation with the reviewers, we have decided that this work will not be considered further for publication by eLife. This decision was reached because ultimately, we felt additional analysis will be required to address some of the key concerns raised, which might not produce sufficiently compelling results. However, should you feel that you are able to conclusively address the main comments made by the reviewers, and in particular the comments I highlight below, I will be happy to consider a resubmission of the manuscript.

We wish to thank the editor and the reviewers for taking their time to read our manuscript thoroughly and for providing insightful comments that helped us to significantly improve the quality of the manuscript.

We read carefully all the comments and we believe that we could conclusively address all the concerns that were raised. We therefore hope that you will find the new version of the manuscript of sufficient quality and interest for a publication in *eLife*.

As our decision was based on our panel discussion, for the sake of transparency I wanted to highlight the key issues that have been raised:1. The ROI approach is too broad, and we want to understand where the information is coming from. This is particularly crucial for the temporal cortex ROI, which encompasses highly heterogenous sub-regions. As you will see below, there was a consensus that the paper contains too many overlapping analyses. With regards to the need for spatial clarity (e.g. searchlight), we agreed in our discussion that this analysis should focus on 2 key results:(i) distances/decoding group differences – where in temporal cortex do they see reduced information content?(ii) group differences in correlation with the visual category RDM (or the categories model) – where in the temporal cortex do they find greater sharing with visual cortex?

Thank you for summarizing those main points. We agree that our temporal cortex ROI was too broad. In line with the idea of streamlining our analytical pipeline (see our response to comment #2), we decided to use the results of our univariate analyses to define smaller ROIs in the occipital and in the temporal cortex for further MVPA (i.e. MVP classification and RSA with different representational models). [Note that is no circularity in using univariate analyses to define ROIs for further MVPA as their goal and principles are separate. Indeed no information about auditory categories is clearly represented at the univariate level in occipital regions while this information is accessible through technique using a distributed approach across voxels]*.* In support of these analyses, we also provide results from a whole brain searchlight approach for the decoding analysis (see Figure 3. Supplemental figure 1), which allow spatial clarification of our results. As you will see, ROI and searchlight analyses converge toward a similar conclusion about which part of the temporal cortex gets reorganized in the blind.

You can find all the information related to the univariate analyses and ROI selection at p. 11-12 (methods section), p. 20-23 (result section) and in Figure 2 (for visualization of the ROIs and the mean β values) and Figure 2 supplemental Figure 1 for the maps with whole brain univariate results.

2. The paper includes too many different analyses which dilute the main findings and it has been our shared view that the balance of main (versus supplementary) analyses should be reconsidered. As you will see, there was some inconsistency across reviewers over which analyses you should focused on, and I believe that ultimately this should be the authors' call. But more careful editing of the Results section is necessary.

Having different analyses converging to similar results was somewhat comforting for us to support the reliability of our results. However, we agree that such redundancy impaired the clarity of our study. Following your suggestions, we now have selected the most insightful analyses and excluded (or placed as supplementary) those more complementary.

In the new version of the paper the main analyses we include are:

1. Using univariate analyses, we were able to isolate a portion of STG in the temporal cortex that is more active in sighted compared to blind subjects and a portion of the ventral occipito-temporal cortex that is more active in blind when compared to sighted subjects during sounds’ listening. Since enhanced-reduced univariate analyses are used in the literature to support “better” processing in the blind temporal cortex (see our introduction about this fallacious status), we decided to go beyond univariate and use MVP-decoding to look at whether sounds encoding was altered.

2. We discovered that in both early and late blind the enhanced coding of sound categories in occipital regions is coupled with lower coding in the temporal regions compared to sighted people. We then asked whether the representation of a specific category of sound was altered in blind people, so we ran a binary decoding analysis on the main 4 categories (human, animal, manipulable objects and big objects/places) which allowed us to observe the decoding of each pair of categories separately. These new MVP-decoding analyses revealed that the representation of the voice category is the one that is most altered in both blind groups. However, as raised by the reviewers, it could be that this alteration in the encoding of voice (reduced in temporal, enhanced in occipital) is due to an alteration of some low-level acoustic aspects typically associated with vocal sounds. To investigate this, we rely on RSA.

3. Using RSA, we investigated which dimension of our stimuli may determine the response properties of the occipital and temporal ROIs. In the temporal cortex, we found that in every group the best model was a “human” model, in which human stimuli were considered similar between themselves and different from all other animate and inanimate stimuli (Figure 4D). Interestingly, we also found that the human model, when compared to other models, showed the highest correlation with the representation of the auditory stimuli in the occipital ROI (Figure 4C) of both our blind groups but not in the SC group. Moreover, the correlation between the occipital ROIs and the human model was significantly stronger in both blind groups when compared to the sighted controls (see Figure 4C and Figure 5). We interpret this partial shift of “human-centric” representation from the temporal to the occipital cortices of blind individuals as a redistribution of computational load across temporal and occipital regions. Crucially, we show that no alteration in the encoding of acoustical features of our sounds (Pitch, HNR) is found in blind people in the temporal and occipital cortices; and that those acoustical features are represented in separate temporal regions (see searchlight analyses in Figure 6 of the manuscript).

In addition we added as supplemental material (Appendix 1):

1. Topographic analysis of VOTC. This test the longstanding question as to whether crossmodal plasticity in late blind is less functionally organized (e.g. follows less the categorical organization of VOTC for vision) than what is observed in early blind people.

3. The results from the LB group are inconclusive and should be interpreted with more caution.

We agree that in the previous version of our article, there was some confusion about the results related to late blind group and about what those data added to the existing scientific literature.

With this new version of our paper, we believe that our data in late blind people are conclusive and add important and novel information on how the onset of blindness impacts the organization of cortical regions coding for the preserved and deprived senses.

Previous studies suggested that late blindness triggers a reorganization of occipital region that is less functionally organized than the one observed in early blindness (Bedny et al., 2012; Collignon et al., 2013; Kanjlia et al., 2019), promoting the idea that crossmodal plasticity in late blindness is more stochastic and epiphenomenal compared to the one observed in early blind people. This is the dominant view in the literature on blindness. However, our results overturn this view by showing functionally specific coding of sound categories that is present in late blind and increased compared to sighted controls. This has broad implications since it supports the idea that the increased representation of sound categories in the VOTC of early and late blind people could be an extension of the intrinsic multisensory categorical organization of the VOTC, that is therefore partially independent from vision in sighted as well (Mattioni et al., 2020; see also Amedi et al., 2002; Ricciardi and Pietrini, 2011; Striem-Amit & Amedi, 2014). Indeed, for such conceptual view to be true, late visual deprivation should maintain or even extend the non-visual coding that is already implemented in the occipital cortex of sighted people. If it was not the case, it would be a serious drawback to the idea that the occipital cortex of blind people maintain is functional organization while enhancing its tuning to the non-visual sense. Our data support this hypothesis, helping to fill this gap in the literature.

We also admit that in the previous version of our paper, the proliferation of analyses presented made it confusing to understand the importance of the results. In the new version of the paper, we now highlight in a clear and straightforward way the relevance of our results in late blind people. Importantly, once we increased the spatial resolution of our analyses (using smaller ROIs coming from independent univariate results and a whole-brain searchlight approach) the results from the LB group become much clearer and straightforward to interpret. Indeed, we have significant group differences and group by region interaction in the decoding data when we compare the LB with SC and, similarly, we have stronger and significant results for the LB/SC comparison also for most of the RSA analyses.

We thank the reviewers and editors for their comments as they were right pointing out that the use of an overly big temporal ROI was partly hindering some important effects/results.

Note that we modified the abstract based on the different analysis pipeline and results that we obtain in the new version of the paper:

“Using a combination of uni- and multi-voxels analyses applied to fMRI data, we comprehensively investigated how early and late-acquired blindness impact on the cortical regions coding for the deprived and the remaining senses. First, we show enhanced univariate response to sounds in part of the occipital cortex of both blind groups that is concomitant to reduced auditory responses in temporal regions. We then reveal that the multivoxel encoding of the “human voice” category (when compared to animals, manipulable objects and big objects or scenes) is reduced in those temporal and enhanced in those occipital regions in both blind groups. Importantly, we additionally show that blindness does not affect the encoding of the acoustic properties of our sounds (e.g. pitch, harmonicity) in occipital and temporal regions but instead selectively alter the categorical coding of the voice category itself. These results suggest a functionally congruent interplay between the reorganization of occipital and temporal regions following visual deprivation, across the lifespan.“

4. Perhaps our biggest issue was the use of stimuli based on visual functional organisation to study auditory representational structure, and compare this categories model to other models which are unlikely to be relevant to auditory cortex (e.g. manipulable objects). To clarify – we do know there are categorical responses in the auditory cortex, but as shown by other groups -they are anatomically constrained and in order to ensure they are indeed object-category responses, you need to better control for low-level auditory features. The fact that the ROIs are bilateral and huge (not only in size, but also in the diversity of selectivity of different subregions of the temporal cortex), combined with a model that does not represent the best response selectivity of the auditory cortex, can result in spurious findings. For example, let's say the best model for the right temporal cortex is pitch-based (very likely based on previous research), whereas for the posterior left is temporal or categorical. When you average them together these two may cancel out and push the results towards a different model or significant difference. We are not sure whether there's a way to solve this issue, but considering the key focus on auditory cortex we feel that there needs to be a better consideration of the native representational features before you can make any strong claims for crossmodal plasticity.

This is, indeed, an important point. When we designed our experiment, we actually closely looked at the literature on auditory functional organization (obviously there is much less on this when compared to vision) and all sound categories included have been previously used to investigate categorical preference to sounds in the temporal cortex. Some of the most prominent studies investigating auditory categorization in the temporal cortex include voices (the temporal voice area; Belin et al., 2000, 2002), music (Haignere et al., 2015; Boebinger et al., 2021) objects and/or tools (e.g. Leaver and Rauschecker, 2010; Lewis et al., 2005, 2006; Murray et al., 2006; Doehrmann et al., 2008), animals (Altmann et al., 2007; Doehrmann et al., 2008; Lewis et al., 2005; Giordano et al., 2013), places and big objects (Giordano et al., 2013; Engel et al., 2009). In particular, the first (and still one of the rare) studies investigating and demonstrating auditory categorization in temporal regions that is partly independent from low-level features using RSA used a stimuli set very similar to ours (Giordano et al., 2013); and we inspired from this seminal study.

In the previous version of our paper, we agree we had not shown empirically that our stimuli (1) are suitable for investigating categorical auditory representation, (2) that are at least partially independent from the representation of low-level auditory properties in the temporal cortex. Actually, we had made these analyses but decided not to include them to not overload the paper but we now realize they are crucial to explain the validity of our stimuli space.

RSA analyses have the advantage to be able to partially dissociate which brain regions code for specific features (e.g., high vs low-level) of our sounds (Giordano et al., 2013). Therefore, we now added further RSA analyses demonstrating a robust categorical coding of our sounds that are at least partially independent from some low-level properties like pitch or Harmonicity (see p 19-20 for the methodological part, p. 31-32for the description of the results and Figure 6 for visualization of the results). This categorical coding of sounds is less topographically clustered than the one observed in VOTC for sight but is robustly expressed in distributed pattern of activity in higher-level temporal regions (Superior Temporal Gyrus; Giordano et al., 2013). In contrast, building a model of pitch representation of our stimuli set using RDM correlated with the brain RDM of the primary auditory cortex (Heschl gyrus). Crucially, no difference was found between blind and sighted groups in the coding of low-level acoustic features (e.g. pitch); while our categorical models correlated with higher-order temporal regions (regressing out the pitch model) more strongly in sighted when compared to our blind groups [please note that we control for the fact that the lower categorical coding is not due to higher pitch coding in the blind in these regions as could be the case with partial regression].

Here follows a more detailed examination of the appropriateness of our sounds to explore how the brain encodes them in both blind and sighted.

We first run a split-half analysis combined with a searchlight approach to show that our auditory stimuli produce a stable pattern of activity in the temporal cortex (Figure 6-supplemental Figure 1). Basically, we split the data in two halves, and we compute in each sphere of the brain a value of stability of the pattern of activity produced by the sounds (i.e. we create for each voxel a matrix including for each stimulus the correlation between the patter of activity that such a stimulus produced with the pattern of activity produced by all other stimuli. In our case it is a 24*24 matrix, since we have 24 sounds in total. Then, we compute the average of the on-diagonal values minus the average of the off-diagonal values and we use the obtained value as the “stability value”). Our data show that in all the groups (i.e. SC, EB and LB) a big portion of the temporal cortex (including the superior temporal gyrus -STG-, part of the middle temporal gyrus – MTG- and the Heschl Gyrus both in the left and in the right hemispheres) show a highly significant stability of the patterns, suggesting that these portions of the temporal cortex has a stable representation of the sounds we selected. Interestingly, we also observed that the split-half correlation is increased in the occipital cortex of both early and late blind groups compared to the sighted and it is, concomitantly, decreased in the temporal cortex of both blind groups (EB and LB) when compared to sighted. A result that goes well in line with our main decoding results.

Then we used RSA with partial correlation to look whether we could segregate the representation of categorical versus low-level auditory properties of our stimuli. To do so, we compute the correlation between the brain representation of our stimuli with either a human, a pitch or a HNR model regressing out the partial correlation shared between these models.

As you can see in the brain maps (see Figure 6), we could segregate a portion of the temporal cortex representing the pitch regressing out human coding (on the right Heschl gyrus as suggested by previous literature and by the reviewers as well), a portion of the right STG representing HNR regressing out human coding and a portion of the temporal cortex representing the human coding regressing out pitch and HNR (bilateral superior lateral temporal region). We now added this analysis and this figure in the main manuscript (see p. 18-19 for the description of the analysis, p. 31-33 for the results and Figure 6).

5. Your main observation for better decoding in the VOTC and worse decoding in the auditory cortex in blind participants is not novel, it has been reported in several papers, but the auditory cortex was usually treated merely as a control site (see Figure 4 in van den Hurk et al., 2017, PNAS; or Figure 4 in Vetter et al., Curr Bio, 2020). We agree that your findings extend and elucidate these original observations, and in my personal opinion the fact that this has been previously observed only strengthens your argument. But there needs to be greater transparency throughout the paper relating to what innovation the paper offers, and give credit to others when it is due. This also extends to the overlap with your previous eLife paper – none of the reviewers could understand if you are re-using your original dataset or replicating your previous findings.

Thank you for raising this important point. We now modified the introduction including more references to previous works that, indeed, already suggested less/more decoding in the occipital/auditory cortex, respectively.

See for instance paragraph 3 in the introduction:

“A few studies reported an increased representation of auditory stimuli in the occipital cortex concomitant to a decreased auditory representation in temporal regions in congenitally blind people (Battal et al., 2021; Dormal et al., 2016, Jiang et al., 2016, Hurk et al., 2017, Vetter et al., 2020). However, these studies did not focus on the link between intramodal and crossmodal reorganizations in blind individuals. For instance, we do not know based on this literature, whether this increased/decreased representation is driven by similar or different features of the auditory stimuli in temporal and occipital regions. Here, using RSA, we explore for the first-time which features of the sounds (acoustic or categorical) are less or more represented in the auditory or visual cortex of blind compared to sighted subjects, respectively”.

Moreover, we now clarify in the methods section the fact that we are re-using the same data collected in early blind people for our previous *ELife* paper. Obviously, the late blind group and additional sighted control subjects are new data:

“All the EB and 17 of the SC subjects were the same participants included in Mattioni et al., 2020. The data from the LB group and from 3 additional SC subjects were never presented before and acquired in the same MRI scanner, with the same protocol and in the same time period (July 2015-April 2016).“

Our study is clearly a direct extension in method and scope of the Mattioni, *eLife* 2020; this is why we decided to submit this paper as an *eLife* “Research Advances” format which is a format for substantial developments that directly build upon a Research Article, published previously by *eLife*. Our data therefore seems particularly suitable for this format of the journal.

Reviewer #1 (Recommendations for the authors):The authors explore the information content (decoding accuracy) and representational structures of sound categories in the auditory cortex of individual with early- and late-onset blindness (EB/LB). They document a slight reduction in information content in auditory cortex in both early and late blinds relative to controls. They also find greater similarity in the representational structure of these sound categories with that of occipital cortex, which has already been shown in their previous work to be greater (more information) and also more similar to visual representation in the EB – this is extended here for the LB. The authors conclude that there's a redistribution of computational load which is not dependent on age of blindness onset. While this interpretation is speculative, it is quite interesting and the evidence, while modest, add novelty and context to the field.A key advantage of the present study is the methods, including detailed accounts of the considerations underlying the various statistical tests which promotes both transparency and education of best statistical practices. However, considering the richness and density of the methods, this can be a double edged sward, and I find that a key disadvantage of the paper is that it is over-processed with multiple analyses, diluting the results of interest. This weakness should be easy to address. Another important advancement, relative to previous research, is the inclusion of the LB group. However, this too introduces a major issue as the effects of interest are quite small, and the study is likely to be underpowered to thoroughly address the impact of blindness onset. This could be addressed with more cautious interpretation of the findings.

Thank you for this evaluation. We have now thoroughly reworked the manuscript to (1) streamline the analytical pipeline, mostly by pruning away redundant analyses (see our response to point #2 of the editor), (2) highlighting better the significance (both theoretical and statistical) of the data collected in the LB group.

We have also rewritten more cautiously our interpretations of the observations.

1. Starting with the abstract and throughout the paper, the authors tend to make causal inferences from associations (e.g. reorganisation in occipital cortex impacts response profile in temporal cortex). But the observed findings in temporal cortex could be driven by different sources altogether (e.g. group differences in perceptual abilities, learning, attention, etc). These multiple drivers, as well as any behavioural differences in performance across groups, should be considered alongside the interpretation of a redistribution of computational load due to occipital cortex reorganisation. Similarly, non-significant results are sometimes taken as evidence for the null (e.g. nonsignificant interaction between EB and LB) and at other times as supporting the hypothesis (e.g. nonsignificant group difference between LB and controls in temporal cortex). Further evidence (e.g. Bayes Factors) should be provided in each case to better inform us about these findings.

We thank the reviewer for these comments. We agree that there could be multiple factors explaining the response profile in the temporal cortex of blind people and this is now discussed in the new version of our paper (see Discussion section p. 35-36).

As we now point out in the discussion, if the different profile of the temporal cortex was driven by group differences in global perceptual abilities, learning or attentional factors, one could expect these factors may impact on the representation of sounds in general, including the representation of low-level acoustical properties. Our new results (not included in previous version and related to our response to point #4 of the editor) show that blindness selectively reduce the representation of the “human model (voice)” in higher-level temporal regions while leaving intact the representation of low-level sound features (Pitch, Harmonicity) and other higher-level models (see RSA correlations with representational models: Figure 4). It is well possible that such alteration in the encoding of voices (compared to other categories and independent from acoustical features) in temporal (reduced in blind) and occipital (enhanced in blind) relates to perceptual learning mechanisms that impact behavior. Some studies have indeed found differences in the way blind and sighted people process voices (Bull et al., 1983; Hölig et al., 2014a, 2014b; Klinge et al., 2010). The relation between such perceptual/behavioral differences and our observation of brain reorganization for voice processing should be explored in future research.

Here is the related section added in the discussion:

“Could such a difference be driven by general factors like group differences in perceptual abilities, learning and/or attention? If the different profile of the temporal cortex between blind and sighted individuals was driven by such general factors, one could have expected a difference not only for the decoding including voice stimuli or for the representation of the human model but also for other stimuli and representational models (e.g. low-level acoustical properties of the sounds). Our results show that this is not the case (see binary decoding including or not including voices: Figure 3B and 3C; and see RSA correlations with representational models: Figure 4). Whether such specific alteration in the representation of voices relates to difference in the way blind and sighted people process voices (Bull et al., 1983; Hölig et al., 2014a, 2014b; Klinge et al., 2010) remains to be explored in future studies”.

We also agree with the reviewer that in the previous version of the paper we had a confused interpretation of non-significant results. Importantly, as mentioned in a previous point of this review, we now have much clearer/simpler results which are more straightforward to interpret. We agree that Bayes Factors is a good addition to interpret non-significant results, notably to determine whether non-significant results support a null hypothesis over a theory, or whether the data are just insensitive (Dienes et al., 2014). However, in the current version of the manuscript we do not interpret anymore nonsignificant results. In relation to the 2 cases of non-significant results mentioned by the reviewer: (1) we do not look anymore at the interaction between EB and LB because the ROIs (based on univariate contrasts with SC) are now different for the 2 groups, therefore we only look at the direct contrast between the EB and the LB in the searchlight approach analyses; (2) the group difference between late blind and controls in the temporal cortex is now significant for the cross-validation analysis (see Figure 3A in the paper). Indeed, as shown in Author response image 1, if we compute the Bayes Factor for this contrast we obtain a factor maxBF_10_=3.436 suggesting that the results support our hypothesis (H_1_).

**Author response image 1. sa2fig1:** Bayes Factor for the MVP results in the temporal ROI LB>SC.

2. Throughout the paper the authors split their control group to 2 (massively overlapping) control sub-groups. This seems unnecessary at best, and at its worst – introduces potential confounds. In particular, to compare between the two blind groups, the authors subtract the mean of each of the sub groups and divide this by the standard deviation of the relevant control sub group. But considering the groups are almost identical in their makeup, this normalisation is essentially taking into consideration the few control participants that were not included in both groups. I believe it makes better sense to add an age regressor and include all participants in the group comparisons. This way the regressor can account more accurately for potential impact of age on the dependent variables and in that respect – the broader the age range (e.g. in the controls, across the two blind groups) the more accurate the regressor. I would therefore encourage the authors to recalculate their group comparisons. Relatedly, I think that a direct comparison between the two blind groups would be valuable.

Thanks for your suggestion. Our choice to use a subset of the SC as SCEB and SCLB groups of controls was driven by the aim of having a matched group of controls for each of our two groups of blinds. However, we believe that the suggestion of the reviewer to add an age regressor is helpful to overcome the age-matching issue. In the new version of the paper, we now include all sighted participants in one group (n=20) that we use as control group for both EB and LB (including the age as regressor of non-interest as suggested by the reviewer).

For the analyses in which it is not possible to add a regressor of non-interest (e.g. permutation analyses), I report in the main paper the results including all the subjects in each group but I repeated them for the SC/LB comparisons including only a subset of the SC that was age matched with the LB. I included this additional data in the supplemental material.

I added this information in the “Participants” section:

“The LB and SC were gender (Χ2 (1,35)=0.03; p=0.87) matched but not age matched (range 25-68 years, mean ± SD: 44.4 ± 11.56 for LB subjects, range 30-63 years, range 2363 years, mean ± SD: 35.1± 8.83 for SC subjects; t(33)=2.70, p=0.01). For this reason, in every parametric test in which we statistically compared the groups we included the Age as regressor of non-interest. For the permutation tests we report in the main paper the results including all control subjects but in the supplemental material we added the results of the same tests including only a subset of 15 sighted subjects age and gender matched with the LB group. We did not find any relevant change in the results when the sighted were all included or when we included only a (age matched) subset of them.”

Note that I did not repeat the analyses for the EB because the group of 20 SC is age and gender matched with the group of 16 EB.

Finally, in the new version of the paper we added the direct comparison of the EB and LB (even if the two groups are not age matched) only in the searchlight analyses (in which we added the age regressor as variable of non-interest). We did not add the direct comparisons of the two blind groups for the ROIs analyses because we now define different occipital and temporal ROIs for the EB and the LB groups based on the univariate groups’ contrasts (EB vs SC and LB vs SC).

3. The "late" blind group ranged acquisition age of 6 to 45. This raises the question – should they be taken as one group? Losing sight in childhood is strikingly different from adulthood, as well as the progression of blindness etc. This adds a level of heterogeneity to this group that should be considered more carefully.

It is true that the late blind group is, by nature, heterogeneous. This is a common issue in our field. Our laboratory now has around 20 years of experience in working with blind people which helped to develop detailed screening procedures about the most important factors that we need to control for. To be included in our study none of the EB participants should have any visual memory and should not have any record of using vision functionally aside from a rude sensitivity to light (e.g. never had shape or color vision). Visual deficits should be present since birth. In contrast, to be included in the LB group, participants should have had functional vision that they used to navigate their environment or recognize people, place and objects and the participants need to have visual memories. In our view, setting a precise age limit about when someone is considered early blind versus late blind is inadequate and we have always advocated to use functional criteria as those described above as the evolution of blindness is very idiosyncratic (Collignon, EBR, 2009). Also, to be included in our sample, the blind person should have no associated neurological and/or psychiatric conditions and should not take chronic psychotropic medication, which limit even more severely the number of late blind that can be enrolled given the fact that they are generally older (and have associated restriction to participate: for instance having had surgeries preventing MRI such as hip prostheses etc.) and sometimes develop a depression associated with late visual loss.

This is likely why the number of studies involving a late blind group are very scarce since this is very difficult to recruit this population for MRI studies, which support the importance of our study given this context. In our study we managed to recruit 15 LB participants which is a very decent number when compared to previous studies involving LB people (Bedny et al., 2011; Burton et al., 2001; Collignon et al., 2013). Actually, 15 is the number of LB participants we could enroll in our study within a recruitment period of approximately 2 years and after screening more than 50 late blind people across Italy (note that those LB participants that went into the MRI came from all over Italy, some did travel more than 600 km and stayed 3 days in the laboratory for this study).

To partially account for the heterogeneity in terms of onset and duration of blindness in the LB group, we computed the correlation between the age of onset and duration of blindness in LB with our main results: the decoding and the RSA human model results (see Author response image 2 and Author response image 3). None of these correlations shows a significant effect suggesting that our decoding and RSA results in LB group cannot be explained by the onset or the duration of blindness.

We are, however, aware, that we have 15 subjects in the LB group, a value that is not optimal for a correlation analysis, therefore we should be cautious in the interpretation of these results.

**Author response image 2. sa2fig2:** Correlation between MVP-accuracies values and Onset & Duration of blindness in LB.

**Author response image 3. sa2fig3:** Correlation between RSA r values for the human model and Onset & Duration of blindness in LB.

4. The ROIs that are used here are massive, and I'm left wondering where in the brain these effects occur. Further refinement of the spatial analysis (e.g. using a searchlight) would greatly improve our physiological understanding of the underlying mechanism.

Our response is linked to our response to the comment #1 of the editor.

I report the same answer here:

“We agree that our temporal cortex ROI was too broad. In line with the idea of streamlining our analytical pipeline, we decided to use the results of our univariate analyses to define smaller ROIs in the occipital and in the temporal cortex for further MVPA (i.e. MVP classification and RSA with different representational models). [Note that is no circularity in using univariate analyses to define ROIs for further MVPA as their goal and principles are separate. Indeed no information about auditory categories is clearly represented at the univariate level in occipital regions while this information is accessible through technique using a distributed approach across voxels].

In support of these analyses, we also provide results from a whole brain searchlight approach for the decoding analysis (following the suggestions), which allow spatial clarification of our results (see Figure 3 supplemental figure 1 in the paper). As you will see ROI and searchlight analyses converge toward a similar conclusion about which part of the temporal cortex gets reorganized in the blind”.

I would recommend to tone down the use of the descriptive 'remarkable' (currently featured 5 times). (This is not a shampoo advertisement :).

No “remarkable” anymore, the manuscript was thoroughly revised to avoid advertising shampoo ;

Page 35 "This result suggests that the representation of auditory categories in the VOTC of early blind subjects is more similar to the representation of the same auditory categories in the temporal cortex of both sighted and early blind participants." – more similar than what?

This sentence has been removed from the new version of the manuscript.

Reviewer #2 (Recommendations for the authors):The study used fMRI to investigate neural representations of sounds from various semantic categories (such as animals, humans, tools, big objects) in the brains of sighted, early blind, and late blind subjects. The authors report better classification of sounds into semantic categories in the early blind group than in the sighted group when the activation of the ventral occipitotemporal cortex (VOTC) is used in the analysis; an opposite effect was found for the superior temporal gyrus. Furthermore, the authors showed that the stimulus features represented in those two regions are more similar in the blind groups than in the sighted group. Based on these results, the authors propose that the absence of visual experience induces network-level reorganization in the brain – that in blind individuals the processing of "auditory categories" is partly delegated to the VOTC.The research question investigated in the study is interesting and the adoption of multivariate analysis methods is timely, as the number of such studies in this research field is still relatively low. The univariate analysis and the classification analysis seem to partly replicate the findings that have been already reported, but the late blind group is added, which can be interesting in some research contexts. The final RSA analysis (Figure 8 and Figure 9) is very interesting and adds to our knowledge about mechanisms of brain reorganization in blindness.The first issue that makes certain aspects of the study hard to evaluate is that the lack of important information about the study design in the Methods section. The authors write that the present study builds on research that is already published, and that they do not want to repeat information that is already there, which is fine. However, they should at least state if the fMRI experimental procedure was identical to the one that they used previously, and if the same or different subjects took part in the two studies. If the experimental procedure was the same, then it is unclear why the selection of new stimulus set was necessary (p. 7-8). If the procedure was different, then the reason for this change and the implications for the present study should be clearly described.

We thank the reviewer for these comments.

We agree that some information about the methods and participants were missing, and we now clarify these points in the paper:

“All the EB and 17 of the SC subjects were the same participants included in Mattioni et al., 2020. Importantly, the LB group and the 3 additional SC subjects were acquired in the same MRI scanner and in the same period (July 2015-April 2016)”.

Moreover, since the procedure was indeed the same as the one used in Mattioni et al., 2020 we now excluded the section related to the stimuli selection since it was unnecessary to repeat it.

This study is clearly a direct extension in method and scope compared to the Mattioni, *eLife* 2020; this is why we decided to submit this paper as an *ELife* “Research Advances” format, a format designed for developments that directly build upon a Research Article, published previously by *eLife*. Our data, therefore, seems particularly suitable for this format of the journal.

The second issue is that, at times, the manuscript is unclear in describing the theoretical background and the findings of the present study. Contrary to what is written in the manuscript, a few studies have already described differences in decoding accuracies in visual and auditory cortices across blind and sighted individuals (e.g. Dormal et al., 2016, Neuroimage; van den Hurk et al., 2017, PNAS; Vetter et al., 2020, Current Biology). Furthermore, the authors use terms "encoding of sounds in the VOTC" or "encoding of auditory categories in the VOTC" – yet, their RSA results seem to suggest the categorical representation in this region, both in blind and in the sighted subjects. The acoustic and categorical claim should be clearly differentiated (can the semantic category be visual or auditory? Do the authors mean that the VOTC in blind individuals capture acoustic properties that differentiate categories of sounds? Clearer definitions of terms such as "auditory category" or "functional relevance" would be helpful in understanding the authors' claims).

This is an important comment, thank you for raising those points. We agree with the reviewer that in the previous version of the paper the theoretical background was missing some important references. We now include the previous studies mentioned by the reviewer, in which the researcher reported alterations in occipital and temporal decoding when comparing early blind and sighted.

See for instance paragraph 3 in the introduction:

“A few studies reported an increased representation of auditory stimuli in the occipital cortex concomitant to a decreased auditory representation in temporal regions in congenitally blind people (Battal et al., 2021; Dormal et al., 2016, Jiang et al., 2016, Hurk et al., 2017, Vetter et al., 2020). However, these studies did not focus on the link between intramodal and crossmodal reorganizations in blind individuals. For instance, we do not know based on this literature, whether this increased/decreased representation is driven by similar or different features of the auditory stimuli in temporal and occipital regions. Here, using RSA, we explore for the first-time which features of the sounds (acoustic or categorical) are less or more represented in the auditory or visual cortex of blind compared to sighted subjects, respectively”.

Moreover, we understand that some terms we used might have created some confusion. For instance, in the introduction we replaced “the encoding of auditory categories” with “the encoding of categories presented in the auditory modality” and in the discussion we also replaced the expression “auditory categories” with “auditory stimuli”. Indeed, when we speak about auditory categories or sound categories we just refer to the modality of presentation of the stimuli and not to the acoustic properties of them (such as the harmonicity or the pitch).

We take the opportunity here to highlight that using representational similarity analyses we were able to segregate the representation of low-level (pitch and harmonicity-to-noise ratio) acoustical properties of our sounds (mostly in the right Heschl Gyrus for pitch and in a specific portion of the right STG for HNR) from the representation of high-level (categorical) properties of the sounds (mostly in STS, see figure 6 in the paper). In addition, we observed that only the categorical dimension of our sounds, and in particular the vocal information, is reorganized following blindness. Our hypothesis is that because there is a link between the representation of specific categories from sound and from vision (e.g. voices and faces allow person recognition), this link might be altered in the blind. On the other hand, no link exists between the encoding of low-level aspects of sound and vision (pitch has no “visual” functional equivalent), therefore, this could explain why the low-level properties of our sounds do not show a reorganization. Those observation represent a step forward in the investigation of concomitant crossmodal and intramodal plasticity following blindness, going beyond what has been reported in previous studies.

Thirdly, certain analytical choices could be better described. The authors write that they used broad VOTC and superior temporal ROIs. However, for the classification analysis, they decided to select only 250 voxels that were the most discriminative for a given comparison in the training data. Could the observed between-group difference be partly explained by the fact that this procedure resulted in the actual ROIs that were systematically different across groups, for the reasons not related to the imbalance across the sensory cortices? Or because the representation of certain stimuli in the blind group is expanded? (e.g., optimal decoding of presented sounds in the auditory cortices requires 300 voxels in this group due to the experience-dependent expansion of sound representation).

We agree with the reviewer that our choice of broad ROIs combined with features (voxels) selection was not ideal (this relates to similar comments #1 from the editor and #4 from R1). In the current version of the paper, we defined smaller ROIs based on univariate group differences, which help in the spatial clarity of our analyses/results. We still use the approach of voxels’ selection in the decoding analysis to run decoding analyses on the same number of voxels in each ROI and Group because these decoding results are inserted in an ANOVA with these factors. Feature selection also optimize decoding analysis by including the most informative voxels (those that contain information); this is standard procedure in the field (e.g. Kriegeskorte et al., 2006; De Martino et al., 2008). However, this voxels selection is made within a much more spatially constrained area. The number of selected voxels (i.e. n=40), is based on the number of voxels of the smaller ROI (i.e. TEMP SC-EB n=42 voxels). In this way, we could select the same number of voxels in each ROI and group.

We now explain in the method section what is the reason driving the number of voxels selected:

“This procedure was implemented in our ROIs: in each cross-validation fold, we first defined from the training data the 40 most discriminative voxels according to our 8 categories (De Martino et al., 2008; Mitchell and Wang, 2007) and then we ran the MVP classification on this subset of voxels in the test data using the parameters described above.

The number of selected voxels (i.e. n=40), is based on the number of voxels of the smaller ROI (i.e. TEMP SC-EB n≈50 voxels). In this way, we could select the same number of voxels in each ROI and group”.

Even if the ROI are smaller, it is still possible that in the bigger ROIs (e.g. occipital ROIs) the selected voxels could differ in the 2 groups. However, we now report as supplemental material the same decoding analyses run with a searchlight approach (see Figure 3 supplemental Figure 1), in which we move a sphere of 100 voxels all over the brain. If the ROI results were strongly impacted by the voxel selection procedure, we should see it in the searchlight analyses but this is not the case.

In addition, to reassure the reviewer that the best voxels selection procedure has not a main impact on the results, we report here the same analysis run without the voxels selection. In other words, we run the classification analyses including for each subject all the voxels in the ROIs. As you can see Author response image 4 the results we obtain with and without voxels selection are very similar. However, we believe that it is methodologically more adequate to keep the voxels selection analysis in our paper, in order to compare statistically results coming from the same number of voxels in each subject and ROI.

**Author response image 4. sa2fig4:** Top: results from MVP-classification analyses using the best voxels selection feature; Bottom: results from MVP-classification analyses including all the voxels in each ROI.

Moreover, the interpretation of the univariate selectivity maps (should be "univariate functional preference maps") in their current form is difficult. The authors report correlation between maps obtained in the auditory modality in the blind group and in the visual modality in the sighted group, but the correlation strength is quite modest and there are clear qualitative differences between the maps in both groups (e.g., no clear preference for animal sounds and a different pattern of preference for human sounds in the blind groups, particularly in the early blind subjects). Inclusion of some sort of noise ceiling (e.g., constructed by correlating visual maps in two sighted groups) would be very helpful in evaluating this analysis.

We realized that this analysis might not fit with the new pipeline of our main manuscript. However, we believe that it is interesting to see that the layout of the topographical map in LB follows a similar layout compared to the one in EB and also to the one observed in sighted for visual stimuli. Therefore, in line with the suggestion to streamline the paper, we decided to move this analysis to supplemental material (see Appendix 1 and Appendix 1-figure 1).

Even if we are not including this analysis as a main one anymore, we found interesting the comments of the reviewer related to this analysis and therefore would like to fully address the point anyway.

We agree with the reviewer that the correlation values between the topographical maps are modest, however the conservative permutation analyses highlight strong significant values (all p<.001 FDR corrected).

As we highlighted also in the minor comment #8 from R1, our permutation procedure for this specific analysis is highly conservative, because we also consider the inherent smoothness/spatial dependencies in the univariate fMRI (more detailed information about the statistical method we applied can be find in the method section of Mattioni et al., 2020).

The reviewer is perfectly right saying that despite robust similarities across groups and modalities in VOTC, there are also some differences between the topographical maps of sighted and blind. We tried to assess these differences in Mattioni et al., 2020 using some more qualitative analysis such as Jaccard similarity (both between subjects of the same groups and between different groups) of the topographical maps and the hierarchical analysis of the dissimilarity matrices extracted in VOTC for each group. Here I report the section of the discussion related to these results from Mattioni et al., 2020:

“Even though the categorical representation of VOTC appears, to a certain degree, immune to input modality and visual experience, there are also several differences emerging from the categorical representation of sight and sounds in the sighted and blind. Previous studies already suggested that intrinsic characteristics of objects belonging to different categories might drive different representations in the VOTC of the blind (Bi et al., 2016; Büchel, 2003; Wang et al., 2015). In line with this idea, the between-groups Jaccard similarity analysis (see Figure 2C) revealed a domain–by– modality interaction, with the big objects and places categories showing the highest degree of similarity between the vision and audition (both in blind and in sighted). In contrast, the lowest topographical consistency between groups was found for the animal category. We found that in the early blind group the number of voxels selective for animals is reduced compared to the other categories (see Figure 2A), suggesting that the animal category is under represented in the VOTC of early blind. Our hierarchical clustering analyses (see Figure 5 and Figure 5—figure supplement 1) also highlight a reduced animate/inanimate division in the EBa group, with the animal and the humans categories not clustering together and the animals being represented more like tools or big objects in the EBa. Interestingly, this is the case in both the categorical representation of VOTC (Figure 5) and the behavioral evaluation of our stimuli made by blind individuals (Figure 5—figure supplement 1). An explanation for this effect could be the different way blind and sighted individuals might have in perceiving and interacting with animals. In fact, if we exclude pets (only 1 out of the six animals we included in this study), sighted individuals normally perceive the animacy of animals (such as bird, donkey, horse etc.) mostly throughout vision (either in real life or in pictures/movies). Blind people, instead, do normally learn the peculiar shape of each animal touching static miniature models of them. Moreover, when blind people hear the sounds of these animals without seeing them, they might combine these sounds with the rest of the environmental sounds, and this is indeed what we see in the behavioral ratings, in which only blind subjects cluster together animals and big environmental sounds. These results therefore reveal that the modality of presentation and/or the visual experience do affect the qualitative structure of the categorical representation in VOTC, and this effect is stronger for some categories (i.e. animals) compared to others (i.e. inanimate).”

We decided to not include these analyses in the current paper to not complexify the paper further (actually “simplification” was a consistent comment across reviewers), since they are not adding much to the main hypothesis. However, out of curiosity and to follow the reviewer’s suggestion, I ran these analyses on the current data, including the LB group.

You find the results in Author response image 5.

We found similar results in LB, as we see in EB. Indeed, in the Jaccard similarity analyses both within groups and between auditory and visual maps (see Author response image 5 central and bottom rows) we observed a similar results as those reported in Mattioni at al., 2020: as for EB, also in the LB group the big objects and places categories show the highest degree of similarity within their own group and also with vision, while the lowest topographical consistency was found for the animal category.

**Author response image 5. sa2fig5:** 

With the hierarchical clustering analysis on the occipital ROI representation (see Author response image 6) we observe a reduced animate/inanimate division also in the LB group, with animal and humans categories not clustering together and the animals being represented closer to tools or big objects in LB, similarly to what we observed in EB.

**Author response image 6. sa2fig6:** 

Beyond these qualitative differences, we believe that our topographical analysis revealed also important similarities. The reviewer for instance mentioned the preference for human stimuli: we isolated this category (voice in audition, face in vision) creating a map only including voxels showing a preference for human stimuli and overlapping these maps across groups (early blind, late blind and sighted control in audition -voice- and sighted control in vision -face-). I include these maps in Author response image 7. As you can see, even if the overlap with the visual map is stronger in the EB, we can still find an overlap also with the late blind and sighted for the auditory map, which is a striking finding in our opinion.

**Author response image 7. sa2fig7:** Human preference in the VOTC of each group and modality.

We also find the idea of calculating a noise ceiling to interpret these data very useful. We calculated the maximal correlation between the subjects within each group (which is one of the most used ways to compute a ceiling also in RSA analyses). We calculated the Spearman correlation of the topographical maps in the visual experiment between sighted subjects (r=0.42) and in the auditory experiment between sighted (r=0.10), early blind (r=0.08) and late blind subjects (r=0.14). These values represent the reliability of the correlational patterns and provides an approximate noise ceiling for the observable correlations between the topographical maps.

As expected, since we are looking at the categorical preference in VOTC, the highest correlation is the one within the sighted subjects in the visual modality. Instead, the representation of sounds in VOTC is more variable than the representations of visual stimuli. This additional information would be helpful in the interpretation of the correlation between groups. Even if they are modest, they, indeed, explain most of the variance under these noise ceilings.

As a final note, we renamed the “topographical selectivity maps” as “topographical univariate functional preference maps” throughout the paper, as suggested by the reviewer.

Finally, what should the reader think about the findings in the late blind group? In the introduction the authors describe the two views of the brain reorganization following late blindness (p. 5; are they mutually exclusive?). However, it seems that the present findings can be well accommodated by either of the views. This aspect of the data should be better discussed. Furthermore, "functional relevance" and "more/less epiphenomenal" are quite unfortunate terms. Based on the current results, the authors cannot claim that the described neural representations in the VOTC are functionally relevant for subjects' behavior.

This point is related to the comment #3 from the editor. We agree that in the previous version of our article, there was some confusion about the results in the late blind group and about what those data added to the scientific literature.

As we now highlight throughout the introduction and discussion of our new version of the paper, we believe that our data in late blind people are conclusive and add important and novel information on how the onset of blindness impacts the organization of cortical regions coding for the preserved and deprived senses. Previous studies suggested that late blindness triggers a reorganization of occipital regions that is less functionally organized than the one observed in early blindness (Bedny et al., 2012; Collignon et al., 2013; Kanjlia et al., 2019), promoting the idea that crossmodal plasticity in late blindness is more stochastic and epiphenomenal compared to the one observed in early blind people. This is the dominant view in the literature on blindness. However, our results overturn this view by showing functionally specific coding of sound categories that is present in late blind and increased compared to sighted controls (see Appendix 1). This has broad implications since it supports the idea that the increased representation of sound categories in the VOTC of early and late blind people could be an extension of the intrinsic multisensory categorical organization of the VOTC, that is therefore partially independent from vision in sighted as well (Mattioni et al., 2020; see also Amedi et al., 2002; Ricciardi and Pietrini, 2011). Indeed, for such conceptual view to be true, late visual deprivation should maintain or even extend the non-visual coding that is already implemented in the occipital cortex of sighted people. If it was the case, it would be a serious drawback to the idea that the occipital cortex possesses a latent representation of sound categories even in the sighted that is scaled-up in early and late blind people (Mattioni et al., 2020; Amedi et al., 2002; Ricciardi and Pietrini, 201).

However, we admit that in the previous version of our paper the results from the late blind group were confusing, notably due to the proliferation of analyses presented. In the new version of the paper, we now highlight in a clear and straightforward way the results about the late blind group and their relevance. Importantly, once we increased the spatial resolution of our analyses (using smaller ROIs coming from independent univariate results and a whole-brain searchlight approach) the results from the LB group become even much clearer and straightforward to interpret. Indeed, we have significant group differences and group by region interaction in the decoding data when we compare the LB with SC and, similarly, we have stronger and significant results for the LB/SC comparison also for most of the (RSA analyses).

Finally, we want to clarify that when we use the terms “functionally organized” and “more/less epiphenomenal” we do not refer to the link with behavior which we cannot claim based on these data. We, instead, refer to the idea that some intrinsic organization might scaffold brain reorganization following blindness. For instance, the preference for a specific category (in the visual domain) might be the constraint that “forces” the same region to prefer the same category presented in a different modality (auditory in our study) in blind subjects. In this sense, the processing of such a region for this specific category in blind for auditory stimuli can be considered not stochastic or epiphenomenal but functionally relevant.

Despite the above-described issues, the reported findings, and particularly the results of the RSA analysis, are an interesting contribution to the debate on the mechanisms of neural plasticity in blindness. The study is so far the most convincing demonstration that blindness affects the representational content of not only the high-level visual cortex (the VOTC), but also the auditory cortex. However, certain aspects of the data analysis and of the claims that are being made can be clarified and improved.

We thank the reviewer for the positive assessment. As we highlighted in our response, we reworked thoroughly both the theoretical background, the data analyses, and their interpretations in the new version of the paper.

Please see below for a couple of additional suggestions.1) What is the noise ceiling in Figure 8? This information seems to be missing.

The noise ceiling of each region and in each group is computed as the correlation between the brain dissimilarity matrix of that ROI between the subjects of the same group (Bracci and Op de Beeck, 2016; Nili et al., 2014). We added this information in the legend of the figure 4, where we plot the noise ceiling:

“Horizontal grey lines show the lower bound of the noise ceiling, which represents the reliability of the correlational patterns and provides an approximate bound of the observable correlations between representational models and neural data (Bracci and Op de Beeck, 2016; Nili et al., 2014).”

2) Page 33: the authors found that there is no correlation between subjects, within the one of the sighted groups, for the VOTC dissimilarity matrices. How to interpret this result? If the VOTC in each sighted subject represents completely different information in the present study, then how can we interpret other results for this region in this group?

We thank the reviewer for pointing that out. We now revised this analysis based on our response to comment #1 of the editor. In figure 3 of the reviewed manuscript, you can now observe the intra-group variability (i.e. the grey lines) in each ROI for each group.

This value is positive for all the groups, including the sighted controls (SC) in audition.

3) The inclusion of the analyses reported in Figures 5-7 in the main text should be better justified – right now, it is unclear if they significantly contribute to the authors' main claims, which creates an impression that the manuscript is overloaded with analyses.

We now excluded these analyses.

**Note from the editor: I've asked R2 for a clarification on their 2nd point in the review, and they added the following:- My comment about the encoding of auditory categories was about clarification, and might be partly related to R3's comment. The STG is sensitive to acoustic features but, at some point, can categorize stimuli into more "semantic" categories, for example human voices. How the STG does this is not clear, but a reasonable (at least to me) assumption is that this "semantic" categorization is at least partly driven by acoustic features – for example, all human voices might have some common temporal and spectral properties and this commonality is captured by certain areas within the STG. Do they propose that, in the blind, the VOTC starts to categorize stimuli based on acoustic features and that this is why this region becomes more similar to the STG (seems to be suggested by "share of computational load" hypothesis and the RSA)? It is not clear to me from the manuscript.

As we replied to the point #2 of Reviewer 2, with RSA and partial correlation we could segregate the portion of STG/STS that represent more high level/categorical/ semantic properties of the sounds from the portions that represents more low-level acoustical properties (i.e. pitch and HNR, see figure 6 in the paper). However, we want to clarify that our hypothesis is the opposite as the one mentioned here by the reviewer. We suggest that low level properties are represented similarly in the brain of sighted and blind subjects. In contrast, we suggest that there is an exchange of information (imbalance) between the semantic representation from auditory (STG) and semantic from visual (VOTC) regions, especially for the “humanness” category. We now clarify our hypothesis throughout the manuscript.

Reviewer #3 (Recommendations for the authors):The paper by Mattioni et al. 2021 studies the effect of blindness on the reorganization of sensory regions of the brain. The paper builds on Mattioni et al. 2020, where the authors used multivariate methods to show categorical representations for sounds in visual regions (ventral-occipito temporal cortex VOTC) in early blind individuals.In the present study, the authors expand this research by addressing two main aims:1) Characterize the effect of onset of blindness on the reorganization of the temporal (auditory) and ventro-occipital (visual) cortex. This is important for our understanding of how sensory experience modulates the organization of sensory cortices, and whether a very early onset of sensory absence has a different effect from a later onset. It provides insights into the importance of developmental sensitive periods, testing whether blindness only results in brain reorganization when occurs early in life, or whether it can cause reorganization at any point, suggesting that sensory experience constantly modulates the functional organization of sensory cortices.2) Understand how sensory experience influences intramodal plasticity, that is, the organization of the sensory cortices that process sensory inputs from the preserved senses. In particular, this paper tries to go beyond simply stating whether responses in the preserved sensory cortex are different between blind and sighted, and aims to understand whether and how the representation of categorical information is modulated by blindness.The paper is ambitious in its goals and design. The authors recruited three groups of participants, including two groups of blind individuals (early and late onset), which is commendable given the challenges of recruiting such populations.The paper builds on the authors previous work, showing that the use of multivariate techniques, in particular representational similarity analysis (RSA), can provide unique insights about crossmodal and intramodal plasticity and the representation of information, addressing gaps in our current understanding of neural reorganization.The main limitations are in the selection of a model and ROIs for evaluating intramodal plasticity and representations in the temporal cortex. The results and conclusions rely very strongly on their choice of an object category model of representation, which does not necessarily represent the selectivity of the temporal cortex or the main dimension of variability of the stimuli set.

We thank the reviewer for the comment.

As already described in our response to point #1 of the editor, and #4 of R1 and #3 of R2, we adopted a different, more transparent and straightforward way to select our (smaller) ROIs for RSA analyses. Moreover, we modified the tested models in order to use models that can better represent the selectivity of the temporal cortex, including models of the acoustic of our sounds (Pitch and HNR) to dissociate higher categorical coding (e.g. it's a voice) from low-level acoustical features that are typically associated with a category (e.g. a voice has a specific frequency range and is more harmonic).

More precisely, we decided to use the results from univariate analyses showing group differences to define smaller ROIs in the occipital and in the temporal cortex for further MVPA (i.e. MVP classification and RSA with different representational models). [Note: There is obviously no overlap in the goal and principles of the univariate and multivariate analyses and therefore no “double dipping” of any kind].

In support of these analyses, we also provide results from a whole brain searchlight approach for the decoding analysis, which allow us to clarify the spatial precision of our results. As you will see, ROI and searchlight analyses converge toward a similar conclusion about which part of the temporal cortex gets reorganized in the blind.

Control of acoustic features in the stimuli is lacking. The first set of analyses reported in the paper, from Figures1-4, assume that the only or most important dimension in which the stimuli are different is their categories into different subsets of objects. Based on the models reported in Figure 8 and information in the methods, many low-level acoustic features vary across categories. These other acoustic features are very likely to drive the responses of the temporal cortex. The choice of the object categorical models seemed to be based on what we know about the VOTC, where these categories and the responses of the VOTC have been characterized in many studies of visual object perception. As the authors mention in the discussion, much less is known about object category organization in the auditory cortex. As such, having stimuli that vary in their spectrotemporal characteristics across categories, variations that are very likely to be represented in the auditory cortex, I am not convinced object category models are the best choice.

This is an important point that we are happy the reviewers raised.

We now actually explore directly these acoustic features, and we show that:

1) Confirming previous studies (Giordano et al., 2013; Leaver and Rauschecker, 2010) we find that acoustic and categorical features of our stimuli are represented in partially distinct regions of the temporal cortex (see Figure 6 in the paper and review-figure 2).

2) Only the categorical (vocal) encoding of our sounds (taking into account acoustic features) is reorganized (reduced correlation between brain and “humanness” model RDMs) in early and late blind people while the acoustic encoding of our sounds (Pitch, HNR) are equally represented in auditory cortex of our sighted and blind groups.

3) VOTC represents the categorical aspect of our sounds but not their acoustic features: it is again selectively this categorical aspect that is enhanced in blind, not the representation of acoustic features. This leads us to conclude that changes in temporal and occipital regions are related since involving similar “higher-level” aspect of sound encoding.

In addition, we also add Author response image 8 in which we show the partial overlap between our ROIs (defined on univariate contrasts) and the results from MVP-decoding.

**Author response image 8. sa2fig8:** Overlap of univariate contrasts and searchlight MVPclassification results.

The authors chose very large ROIs to conduct their analyses. This is problematic, because it is forcing a single outcome (in terms of selectivity, model fitness, classification, etc) from regions that are likely to have different selectivity. This is particularly problematic for the temporal cortex, where the selectivity of anterior and posterior regions varies significantly. Furthermore, averaging right and left ROIs is also problematic for the analysis of the superior temporal cortex, where the right and left have different selectivity for temporal and spectral processing, respectively.

We agree with this point. We now define smaller ROIs constrained by univariate results, and we also include searchlight results (see our response to point #1 of the editor- see Author response image 8).

In their conclusions, the authors suggest that specialization for other senses in regions usually considered 'unisensory' is what allows crossmodal plasticity in cases of sensory deprivation. For example, cortical regions typically considered to be 'visual' areas also show some specificity for processing auditory information, and this specialization is the foundation of crossmodal plasticity effects. This is supported by a similarity in the representation of categories of sounds and images in the VOTC in sighted and blind individuals (both reported here and in Mattioni et al., 2020). However, it is difficult to conclude whether this is also the case in late blind, because even though there is a similar trend a trend, some of the differences do not reach statistical significance.Furthermore, results from the temporal cortex do not show the same selectivity for auditory and visual stimuli. However, this could be due to the author's choice of a model that does not best represent the selectivity of such cortex, and alternative models should be tested to support this conclusion.

This is a great point. It is indeed true that based on the results of our 2 studies, we suggest in our conclusion that a region that is typically considered unisensory such as the visual ventral pathway might also represent information from other senses in a format that is partially aligned with the one used to represent visual information. And that this alignment is the scaffolding for our observation of functionally specific crossmodal extension in the representation of sounds in the VOTC of blind people.

In the previous version of the paper, the results of the LB group were indeed confusing, likely due to the overly complex stream of data analyses steps we presented and our choice of overly large ROIs. In the current version, based on the comments from the editor and reviewers altogether, we changed the definition of our ROIs and selected the most insightful analyses to be presented in the paper. As the reviewer will see, we now have clearer data also for the LB group and we can interpret the results in a more straightforward way. This point is highly related to the comments #3 from the editor and #5 from reviewer 2. As I highlighted already in the response at those comments: we believe that our data in late blind people are conclusive and add important and novel information on how the onset of blindness impacts the organization of cortical regions coding for the preserved and deprived senses.

Previous studies suggested that late blindness triggers a reorganization of occipital region that is less functionally organized than the one observed in early blindness (Bedny et al., 2012; Collignon et al., 2013; Kanjlia et al., 2019), promoting the idea that crossmodal plasticity in late blindness is more stochastic and epiphenomenal compared to the one observed in early blind people. This is the dominant view in the literature on blindness. However, our results overturn this view by showing functionally specific coding of sound categories that is present in late blind and increased compared to sighted controls. This has broad implications since it supports the idea that the increased representation of sound categories in the VOTC of early and late blind people could be an extension of the intrinsic multisensory categorical organization of the VOTC, that is therefore partially independent from vision in sighted as well (Mattioni et al., 2020; see also Amedi et al., 2002; Ricciardi and Pietrini, 2011). Indeed, for such conceptual view to be true, late visual deprivation should maintain or even extend the non-visual coding that is already implemented in the occipital cortex of sighted people. If it was not the case, it would be a serious drawback to the idea that the occipital cortex of blind people maintain is functional organizing while enhancing its tuning to the non-visual sense. Our data support this hypothesis, helping to fill this gap in the literature.

We admit that in the previous version of our paper, the proliferation of analyses presented made it confusing to understand the importance of the results. In the new version of the paper, we now highlight in a clear and straightforward way the relevance of our results in late blind people. Importantly, once we increased the spatial resolution of our analyses (using smaller ROIs coming from independent univariate results and a whole-brain searchlight approach) the results from the LB group become even much clearer and straightforward to interpret. Indeed, we have significant group differences and group by region interaction in the decoding data when we compare the LB with SC and, similarly, we have stronger and significant results for the LB/SC comparison also for most of the RSA analyses.

These are my main comments about improvements to the manuscript:1) The authors conducted a remarkably challenging and ambitious research, with a lengthy and complex set of analyses. I suggest guiding the reader through the relevance of each of the analyses, and reconsider whether they all add to the conclusions, or whether some are redundant. I suggest highlighting from the beginning the most important and innovative results. In my view, those are the RSA results, examining the fitness of different models, and looking at correlations between groups and ROIs.Some small things like adding titles or descriptions within figures (not only in the legends), spelling out some of the acronyms in the titles, will also help the reader in quickly understanding the aim of the analysis and the differences between figures.

Thank you for the positive evaluation and clear suggestions. We now selected a subset of the analyses to be included in the paper, excluding the redundant and unnecessary ones and focusing on those suggested. We agree with the reviewer that the RSA results are the more impacting ones and they now take the central place.

In the new version of the paper the main analyses we include are:

1. Using univariate analyses, we were able to isolate a portion of STG in the temporal cortex that is more active in sighted compared to blind subjects and a portion of the ventral occipito-temporal cortex that is more active in blind when compared to sighted subjects during sounds’ listening. Since enhanced-reduced univariate analyses are used in the literature to support “better” processing in the blind temporal cortex (see our introduction about this fallacious status), we decided to go beyond univariate and use MVP-decoding to look at whether sounds encoding was altered.

2. We discovered that in both early and late blind the enhanced coding of sound categories in occipital regions is coupled with lower coding in the temporal regions compared to sighted people. We then asked whether the representation of a specific category of sound was altered in blind people, so we ran a binary decoding analysis on the main 4 categories (human, animal, manipulable objects and big objects/places) which allowed us to observe the decoding of each pair of categories separately. These new MVP-decoding analyses revealed that the representation of the voice category is the one that is most altered in both blind groups. However, as raised by the reviewers, it could be that this alteration in the encoding of voice (reduced in temporal, enhanced in occipital) is due to an alteration of some low-level acoustic aspects typically associated with vocal sounds. To investigate this, we rely on RSA.

3. Using RSA, we investigated which dimension of our stimuli may determine the response properties of the occipital and temporal ROIs. In the temporal cortex, we found that in every group the best model was a “human” model, in which human stimuli were considered similar between themselves and different from all other animate and inanimate stimuli (Figure 4D). Interestingly, we also found that the human model, when compared to other models, showed the highest correlation with the representation of the auditory stimuli in the occipital ROI (Figure 4C) of both our blind groups but not in the SC group. Moreover, the correlation between the occipital ROIs and the human model was significantly stronger in both blind groups when compared to the sighted controls (see Figure 4C and Figure 5). We interpret this partial shift of “human-centric” representation from the temporal to the occipital cortices of blind individuals as a redistribution of computational load across temporal and occipital regions. Crucially, we show that no alteration in the encoding of acoustical features of our sounds (Pitch, HNR) is found in blind people in the temporal and occipital cortices; and that those acoustical features are represented in separate temporal regions (see searchlight analyses in Figure 6 of the manuscript).

In addition we added as supplemental material (Appendix 1):

1. Topographic analysis of VOTC. This test the longstanding question as to whether crossmodal plasticity in late blind is less functionally organized (e.g. follows less the categorical organization of VOTC for vision) than what is observed in early blind people.

Finally, we also thank the reviewer for the suggestions related to the figures. We now try to use less acronyms and to add more titles and description within figures to enhance their clarity.

2) I suggest to remodel the Results section. One could first address the question the authors mention on page 51 of their discussion: 'Which dimension of our stimuli may determine the response properties of the temporal ROI'. I suggest starting the Results section showing the analysis reported in figure 8 and then use the best models to conduct the rest of the analysis, including correlations between groups and ROIs, as well as the topographical selectivity maps (I understand this will not be possible for models that are not categorical).

We thank the reviewer for these suggestions. We agree that the RSA results examining representational models needed to be emphasized and this is what we do in the current version of the paper. We also run further statistical analyses only on the results from the best model (i.e. the human model).

However, we decided to keep the decoding analyses because they represent preliminary logical analytical steps on which we build to arrive to the RSA results (from univariate preference, to decoding information to more detailed brain representation of the different features of the stimuli). See also next comment for further details on this.

3) The MVPA analysis is also based on an object category representation. It is not clear what this analysis adds to the RSA analysis, and again, it is assuming object category is an important dimension. I will recommend removing this analysis from the paper.

We agree that the complementarity between the decoding and RSA analyses were not clear enough. We believe that the decoding analyses are important to first detect whether early and late blindness alters the representation of our sound categories in part of the occipital and temporal cortices. We indeed assumed our sound categories were an important dimension for several reasons: (1) it is the dimension used by default by sighted and blind people when asked to evaluate the similarity between each pair of sounds (see Author response image 9); (2) we knew from our previous study that sound categories and not lowlevel sound features is the dimension that is overly expressed in the occipital cortex of congenitally blind subjects.

**Author response image 9. sa2fig9:** Dissimilarity matrices resulted from the behavioral similarity rating of each sound pairwise.

What our new study hypothesized is that such enhancement in occipital region may impact the coding of those dimension known to also be encoded in higher-order temporal regions (Giordano et al., 2013). We obviously introduced in our paper a new group of late blind to look at the impact of blindness onset on those reorganization process.

MVP-Decoding analyses tell us that some occipital regions show enhanced decoding, and some temporal regions show decreased decoding of sound categories in both blind groups. In addition, we observed that the voice category was the one mostly driving the difference between the representation in the occipital and temporal cortices of sighted vs blind subjects.

As recommended by the reviewer, we now look at which features of our stimuli space (low-level vs categorical) are less well represented in those reorganized regions, therefore going beyond saying there is a reduced representation of our sound categories. As rightly pointed out by the reviewer, the alteration in the decoding of our sound categories could be explained either by some higher-level representation but also because of acoustic features specific to each category. And this is indeed to address this point that we complement our decoding analyses with RSA. As described in our response to point #4 of the editor, we first show how our stimuli are well suited to address such question by showing, in all groups, that models of some acoustic features of our sounds (pitch, HNR) correlate more with the representational structure implemented in Heschl gyrus while the categorical and Human models correlate more with the representational structure of higher-order temporal regions (e.g. STS) (see figure 6). Then, and importantly, we show that it is only the representational structure of our Human model that is reduced in both LB and EB groups in higher temporal regions, but not the encoding of low-level acoustic features which is preserved in the blind groups (see Figure 4 and Figure 5). This is interesting to us since this relates to the reversed group difference observed in some occipital regions where we find enhanced representation of the Human model in blind but no representation of acoustic models in occipital regions.

Why is that important? It shows for the first time that acoustic features of sound processing are not altered in the auditory region and not represented in the occipital cortex of early and late blind people. In contrast, the representation of a higher-level category “Human/Voice” is reduced in temporal regions and enhanced in occipital regions. What does that suggest? We believe this suggests that blindness triggers a scaling up of a “latent”, “amodal” categorical “human-centric” representation in the occipital cortex. This is filling in an important gap in the literature about how changes in temporal and occipital cortices relate to each other in case of early and late visual deprivation, providing a comprehensive view on the way plasticity expresses following blindness.

4) The authors used very large ROIs for their analysis, and the main issues of this have been explained in the public review. There are several things that can be done to improve this without acquiring more data:1) analyze right and left ROIs separately (it will not solve all the problems, but it will be a significant improvement);2) do a searchlight analysis;3) use ROIs from the Destrieux Atlas (instead of the Desikan-Killiany), and either correct for multiple comparisons or focus on specific temporal regions, such as PT or posterior temporal regions.You can always use one of your functional runs as a functional localizer, but of course, this will significantly reduce the amount of data in your analysis.

We thank the reviewer for these useful suggestions. As described at length in our response (see our response to point #1 of the editor), we now define our ROIs using results from univariate analyses and we support these analyses with the results from searchlight analyses as well, leading to the definition of much more discrete ROIs in temporal and occipital regions.

[Editors’ note: what follows is the authors’ response to the second round of review.]

Essential revisions:As you will see below, the reviewers were generally happy with your revisions in response to the original reviews. However, two new issues have emerged that in particular require further substantial revisions.1) Circular analysis. The reviewers have had an extensive discussion on whether or not your ROI selection criteria might have potentially biased the group results. To cut a long discussion short, we acknowledge that the multivariate analysis is done between categories within individual subjects, whiles the ROI is defined based on the univariate group differences. However, the independence of the classifier from the univariate results is not guaranteed, and we could come up with plausible scenarios under which your ROI selection criteria artificially inflates the classifier group differences. For this reason, we think that at the very least you will need to repeat the ROI definition using a leave-one-out approach (that is, while excluding the subject on which the classification analysis is carried out). We expect that this will not substantially affect your results, but will avoid the issue of circularity.

We agree with this point, and we now select our ROIs using a leave-onesubject-out approach.

In this new version of the manuscript, we defined for each subject the occipital and temporal ROIs using the univariate contrasts between groups, excluding the subject itself from the contrast (E.g. for the EB1 the occipital ROI is defined as the contrast [all EB but EB1)> (all SC]). We then perform decoding and RSA in this ROI that has been defined without the subject itself.

You find the description of this method in the section ROI definition (pp 25).

“Importantly, to avoid any form of circularity, we applied a leave-one-subjectout approach to define individual ROI: for each subject we run the univariate contrasts excluding the subject himself/herself from the analysis (e.g. for the EB1 the occipital ROI is defined as the contrast [all EB but EB1)> (all SC])”.

As anticipated by the reviewers, using the new ROIs had no significant impact on our results neither for the β extraction, nor for the decoding or for the RSA analyses. See the response to the individual reviewer’s points for detailed comments about the results.

2) The new focus on the human model is not well motivated by your experimental design (or your previous study), and at worst could be taken as HARKing. We are more than happy to discuss with you any potential solutions to this issue, but a straightforward one will be to include the full analysis from the original manuscript.

We agree that our experimental design was not made to test the human model specifically. Our decision to focus on this model (vs low-level auditory models) in the last version of the study was driven by our binary decoding results that suggested a central role of the human sounds in driving differences between sighted and blind groups. We started the study with no a priori of which sound categories might be mostly driving differences between blind and sighted subjects in occipital and temporal regions. When observing from the binary decoding including all pairs of categories that the difference between sighted and blind subjects was mostly driven by the decoding of the human category vs all other categories (as we reported in the figure 3B and 3C in the previous version of the manuscript), we thought that it was legit to focus on this category in subsequent RSA analyses and contrast a categorical model of human voices with acoustic models to test whether the reorganization is mostly driven by the categorical aspect of voices vs other categories or was mostly linked to the specific acoustic features linked to voices (e.g. pitch or harmonicity). Our intention was obviously not to hark but to base our analytical strategy building on our analytical steps (here RSA decision was based on decoding results since RSA can complement decoding by testing what information may explain the decoding- here low vs high dimensions of our auditory stimuli).

That being said, we understand that our study might be more comprehensive if we add multiple categorical models in the RSA analyses. Therefore, as recommended by the editor, we went back to the version of the RSA analysis that was included in the original version of the paper. To be coherent across analyses, we now use the new ROIs defined with the leave-one-subject-out approach. These changes did not impact our main conclusion that the human model is the one driving the reorganization in temporal and occipital regions of blind (early and late) when compared to sighted people (see figure 4B and 5). See the response to reviewer’s comments #2 for R1 and #2 for R2 for further details related to the RSA analysis.

Since now we introduced a more complete set of models in the RSA analysis, we believe that the binary decoding analysis is not anymore needed, especially in light of the main suggestion in the first round of review to streamline the manuscript by reducing the number of analyses reported if they are redundant. Indeed, the binary decoding it is not revealing any additional information to the 8-way decoding + RSA encoding analysis with multiple models.

Reviewer #1:The authors have done a fantastic job thoroughly addressing all of my comments from the previous submission. The paper is much easier to read now, and for this reason, its novelty and impact shines even brighter.Unfortunately, the revised version of the manuscript raises a few key methodological and conceptual issues concerning circularity that will need to be ironed out. I hope that those could be addressed with further revisions. Please note that I've restricted my comments to any changes that have been made to the original submission.ROI selectionWhile I understand the conceptual motivation to focus on the areas where there are noted group differences, I'm puzzled by the methodological implementation. ROI selection was based on group differences in activity for the main stimulus. Clearly, greater activity will lead to greater decoding abilities (because there's less information/signal in the control group – see figure 2). So when the same data is used for both defining the ROI and running the decoding analysis, this seems entirely circular to me. To overcome this circularity, the analysis requires a leave on out/split half approach.

As you can read in the response to the point #1 of the editor, we now apply a leave-one-subject-out approach to define the ROIs.

Encoding analysisWhile I found the narrative of teasing apart high level versus low level contributions appealing, i could not understand why 'humanness' was chosen as the high level model. The study was clearly not designed to address this question, as the categories are not equally distributed between human and non human sounds. The study was designed based on categorical similarities/differences, as clearly indicated in the colour code of Figure 1, and this should be the competing model to the low-level ones. Alternatively, high-level representational structure is often derived by the experiential self report of participants.

As mentioned in our response to the point #2 of the editor, we now went back to the original version of this analysis in which we included a set of different categorical and low-level models. Among these models we also include a behavioral model, which is based on the experiential self-report of participants (as suggested by the reviewer and in line with what we did in Mattioni et al., 2020). You can find the description of this analysis in the method section (pp 31/32):

“First of all, we built several representational models (see Figure 5A) based on different categorical ways of clustering the stimuli or on specific acoustic features of the sounds (computed using Praat, https://praat.en.softonic.com/mac).

Five models are based on high level properties of the stimuli (models from 1 to 5) and 2 models are based on low level properties of the sounds (models from 6 to 7) for a total of 7 representational models (See Figure 5A and 5B to visualize the complete set of models and the correlation between them):

1. Behavioral model: it is based on the subject’s ratings of similarity, which were based on categorical features. We included one behavioral model for each group.

2. Human model: it is a combination of a model that assumes that the human categories cluster together and all other categories create a second cluster and a model that assumes that the human categories cluster together and all other categories are different from humans and between themselves (Contini et al., 2020; Spriet et al., 2022).

3. Animal model: it is a combination of a model that assumes that the animal categories cluster together and all other categories create a second cluster and a model that assumes that the animals categories cluster together and all other categories are different from humans and between themselves.

4. Manipulable model: it is a combination of a model that assumes that the manipulable categories cluster together and all other categories create a second cluster and a model that assumes that the manipulable categories cluster together and all other categories are different from humans and between themselves.

5. Big & Place model: it is a combination of a model that assumes that the big & llace model categories cluster together and all other categories create a second cluster and a model that assumes that the big & place model categories cluster together and all other categories are different from humans and between themselves.

6. Harmonicity-to-noise (HNR) ratio model: the HNR represents the degree of acoustic periodicity of a sound.

7. Pitch model: the pitch, calculated with the autocorrelation method (see Mattioni et al. 2020), represents the measure of temporal regularity of the sound and corresponds to the perceived frequency content of the stimulus.”

As a side note, our former decision to focus on the humanness model (vs lowlevel auditory models) in the last version of the study was driven by our binary decoding results (done before RSA analyses) that suggested a central role of the human sounds in driving differences between sighted and blind groups. As rightly mentioned by the reviewer, we started the study with no a priori of which sound categories might be mostly driving differences between blind and sighted subjects in occipital and temporal regions. We observed from the binary decoding including all pairs of categories that the difference between sighted and blind subjects was mostly driven by the decoding of the human category vs all other categories (as we reported in figure 3B and 3C in the previous version of the manuscript). We therefore thought that it was legit to focus on this category in subsequent RSA analyses and contrast a categorical model of human voices with acoustic models to test whether the reorganization is mostly driven by the categorical aspect of voices vs other categories or was mostly linked to the specific acoustical features linked to voices (e.g. pitch or harmonicity).

However, as mentioned above, for the sake of completeness and to streamline our analytical pipeline, we now went back to the older version of our RSA analyses including all categorical models as suggested by the editor. It is important that in any cases, the results indeed suggest that it is the representation of voices that is mainly different across groups in occipital and temporal regions, at least at the multivariate level (see our response to point #3).

Reading the results I realise the authors observed a greater univariate group difference in the human categories and this has likely drove the decision to use the human model in further analysis. But I'd again argue here for circularity (see above) and ask that this is addressed in the analysis.

We respectfully think that there is a misunderstanding here. At the univariate level we did not find a greater difference between groups for the human category. This is shown in Figure 2B and in the Supplementary File 3 were we report the β values for each ROI in each group. From this graph one can see that the difference between sighted and blind individuals in each ROI is driven by all the categories and not by one category specifically. We tested this statistically and we did not find any interaction group*region in none of the ROIs. You can find the results from the ANOVAs in the section *Results – Betas extraction* (pp. 7-8).

Nevertheless, compared to the previous version of the manuscript we now run the RSA analysis in the new ROIs selected using a leave-one-subject-out approach (see also our response to point #1 of the editor). As you can see in figure 4B and in figure 5, the results did not significantly change.

Key hypothesisIn the introduction, the authors set up the main motivation of the current study: "Would the same categorical representation be the one that could be reorganized in the temporal cortex of these blind individuals? If true this would speak up for an interplay between the features that are reorganized in the temporal and occipital cortices of visually deprived people". Based on this interesting framework, the representational structure of sounds in OTC and TC is shared. But the key analysis – a group comparison of the correlation across the RDMs of the two brain areas is not shown to us.

Thank you for this comment. This analysis was there in the original version of the paper, but we now excluded it to streamline the paper based on previous suggestions of the reviewers; indeed one major comment in the last round was to reduce the number of analyses we present. However, as the reviewer is suggesting here, this could be a crucial analysis to bring back to support our hypothesis.

Therefore, we now added back this analysis in the new version of the paper. We correlated the DSMs from the occipital and temporal ROIs of each subject (we used the new ROIs selected with the leave-one-subject-out approach; see our response to point #1 of the editor). Here follows the description of the analysis that one can find in the method section at pp. 17:

“RSA – Correlation between occipital and temporal ROIs in each subject and group.

When the sounds of our 8 categories are presented, brain regions create a representation of these sounds, considering some categories more similar and others more different. Would visual deprivation have an impact on the structure of representation for sound categories in the occipital and temporal regions? Our hypothesis was that the similarity between the representation of the 8 sound categories between temporal and occipital regions was enhanced in blind individuals compared to their sighted controls. To test this hypothesis, we compared the correlation between the DSMs of the occipital and temporal ROIs in each group.

In each individual we computed the Spearman’s correlation between the occipital and temporal DSMs. We then averaged the values across subjects from the same group to have a mean value per group (Figure 4A).

For statistical analysis we followed the procedure suggested by Kriegeskorte and collaborators (2008). For each group, the statistical difference from zero was determined using permutation test (10000 iterations), building a null distribution for these correlation values by computing them after randomly shuffling the labels of the matrices. Similarly, the statistical difference between groups was assessed using permutation test (10000 iterations) building a null distribution for these correlation values by computing them after randomly shuffling the group labels. The p-values are reported after false discovery rate (FDR) correction (Benjamini and Hochberg, 1995)”.

Here is the description of the results (see also pp. 29-30):

“RSA-Correlation between the representational structure of Occipital and Temporal ROIs.

The results of this analysis are represented in figure 4A. We looked at whether the representation of the 8 sound categories shares any similarity between the occipital and the temporal parcels within each blind and sighted subject, with particular interest at group differences. The permutation test revealed a significant correlation between the representational structure of occipital ROI and the representational structure of the temporal region only in blind groups (EB: r=0.12, p<0.01; LB: r=0.14, p<0.01), but not in SC group (r=0.02 in both ROIs). When we look at the differences of correlations values between groups, we found a significant difference between the EB and the SC groups (p<0.01, FDR corrected), highlighting an increased similarity between the occipital and the temporal DSMs in the EB when compared to the SC group (Figure 4A). The difference between the LB and the SC (Figure 4A) was also significantly different (p<0.001, FDR corrected), showing an increased similarity between the occipital and the temporal DSMs in the LB when compared to the SC group (Figure 4A)”.

We thank the reviewer again as we think bringing back this analysis increases the significance of our study.

Open questionThe fact that the low-level auditory models did not capture significant variance in the temporal cortex (and in fact seemed to perform similarly, if not better in the visual cortex of sighted controls) calls for a more serious characterisation of the brain area under investigation in control participants.

It is indeed true that in the RSA analysis in the ROIs, the low-level (pitch and HNR) models did not show a significant correlation.

However, we see in the searchlight analysis (figure 6) that these two acoustic models show a significant correlation with specific portions of the temporal cortex, mainly the primary auditory / Helsh cortex (that does not overlap with our regions of interest). These results from the searchlight RSA analysis demonstrate the possibility of capturing low-level acoustic representation of our auditory stimuli.

The fact that these models are not explaining the representation in the temporal ROI could be explained at least in two ways: either this region is more highlevel and therefore its representation is better explained by high-level/categorical models (e.g. behavioral or human models), or there are other low-level acoustic features represented there that we did not test. We could in theory test numerous models with RSA, but we would then have to deal with the correction for multiple comparisons. For this reason, we decided to only test 2 low-level acoustic models (pitch and HNR) which previous literature pointed out as two highly represented features in the temporal cortex (Giordano et al., 2013, Leaver et al., 2009).

In the new version of this analysis (RSA in the new ROIs with high/low-level models) we now include five high-level models (i.e. behavioral, human, manipulable, places and animals models) and two low-level modes (i.e. pitch, HNR).

Figure 3C seems to me circular to Figures 2 and 3B, and I suggest removing it.

In the previous version of the paper, we ran a binary decoding analysis to better understand the role of each category in driving the difference between sighted and blind groups. The results from this analysis highlighted a crucial role of the human (voice) sounds and these results were depicted in old figure 3B and 3C. Starting from this observation/result, we used RSA analysis to better investigate the representation of human sounds (including only the human model and the 2 low-level auditory models in the RSA analysis).

Since now we introduced a more complete set of models in the RSA analysis (see our response to point 2 of the editor and point 2 of the reviewer), we believe that the binary decoding analysis is not anymore needed. Indeed, it is not really revealing any additional information to the 8-way decoding + RSA encoding analysis with multiple models.

Based on this reasoning, and in order to streamline the paper with only key analyses (one of the main suggestion of round 1), we decided to exclude the binary decoding analysis and therefore removed the figure 3C from the paper.

In the 'stability analysis' for the searchlight analysis, I couldn't quite understand why not run the same decoding analysis used for the main analysis (Figure 3)? As a side note – do the areas identified here in OTC actually overlap spatially with the ROI used in the main analysis?

We relied on the split-half analysis to evaluate the reliability of our data, since this analysis specifically look at how stable are the patterns of activity in the two different halves of the data. We however agree with the reviewer that the decoding analysis is a good proxy for the stability of the patterns. In contrast to the split half-analysis, the 8-way decoding looks at the stability of the categorical representation (all stimuli within each category mixed); while with the split-half we are looking at the stability of the patterns of the single items. For these reasons, we think that the split-half should be preferred testing the reliability of pattern of activity across instances of presentation of our different stimuli.

In figure 3-supplemental figure 1, we also reported the decoding analysis combined with a searchlight approach.

Related to the question about the overlap, I report in Author response image 10 which I show results from both the univariate contrasts and the 8way-decoding in the searchlight analyses. In all the cases we can find a reliable degree of overlap.

**Author response image 10. sa2fig10:** 

As a side point, is it of any relevance/interest that the temporal ROI is in a different hemisphere for each group?

At a group level and at a stringent correction for multiple comparisons there is a clear hemispheric difference between the two groups of blind subjects. However, we couldn’t find any specific information related to this in previous literature and it is quite difficult to interpret this difference.

Importantly, this laterality difference in the temporal cortex does not emerge when we directly compare the EB vs LB groups (even at a very lenient statistical threshold of p<0.01unc).

Actually, when the statistical correction is less stringent (i.e. p<0.01unc.) we can observe a bilateral activity in both group comparisons (SC>EB and SC>LB), suggesting that this hemispheric difference is apparent only at a specific (stringent) level of statistical correction, but somehow disappear at a more lenient threshold.

Altogether, this suggests that there is not a reliable hemispheric difference in the reorganization of temporal regions in the EB and LB and that we should probably be cautious in interpreting our temporal ROIs laterality. For these reasons we decided not to discuss this point in the paper.

I was missing a direct comparison between the two blind groups to really bring home the message that they are not different from one another. Here, of course, some care should be taken into demonstrating evidence to support the null hypothesis (e.g. BF).

We also compared at the univariate level the EB vs the LB group (see Figure 2 supplemental figure 1 EB>LB & LB>EB) but we did not find any difference within the occipital and temporal cortices.

Our ROIs are different between EB and LB subjects, since they are based on univariate difference between EB and LB with the sighted group separately. We therefore never directly compared the EB with the LB in the ROIs analyses. However, the reviewer can find a direct comparison in all the supplemental analyses run with a searchlight approach: see Figure 3-supplemental figure 1 for the 8-way decoding analysis and figure 6 -supplemental figure 1 for the split-half analysis. Only the RSA analysis was run exclusively in ROIs, therefore we did not directly compare the EB and LB groups for this analysis.

Reviewer #2:Thank you very much to the authors for their effort in reviewing their paper. It has improved significantly, and the aims and rationale for the different analyses are much clearer.The paper relies strongly on the results obtained with the RSA and MVPA analysis, but I have concerns about the circularity in the definition of ROIs, which bring to question the reliability of the results. I disagree that the definition of the ROI is not circular. The authors define the ROIs on differences across groups, and then use these ROIs to show that differences in classification across groups. This is circular, as in both cases the authors are looking at group differences. For example, in the MVPA analysis, a difference in intensity will also result in categorical classification differences. This, combined with the fact that differences between groups in the searchlight analysis are not significant at corrected level, puts in doubt the claim about reduced classification in the temporal cortex in blind individuals.

We thank the reviewer for highlighting this point.

As described in the response to the point #1 of the editor, and to point #1 of reviewer 1, we now applied the suggestion of the reviewers and editor, that is to select our ROIs using a leave-one-subject-out approach.

In this new version of the paper, for each subject we defined the occipital and temporal ROIs for further multivariate analyses using the univariate contrasts between groups, excluding the subject itself from the contrast (E.g. for the EB1 the occipital ROI is defined as the contrast [EB (excluding EB1)> all SC]).

In addition, it is not clear how the authors went from a variety of models in their original manuscript, to the three models displayed in Figure 4. It is difficult to believe that these models capture the full variability of their stimuli. Take for example the results of Figure 6, where the "Human" model is the one that captures the best activity across most of the STC. It is known that STC does not only code 'human' vs 'non-human', which highlights that there is information missing in the models used.

We agree with the reviewer that the human model is not capturing the full variability of the stimuli in the temporal ROI.

In the previous version of the paper, we were using a binary decoding analysis to better investigate the role of each category in driving the difference between the blind and sighted groups. That analysis highlighted a crucial role of the human sounds in explaining the group differences. Our decision of only including the human model in the RSA analysis was based on these results.

However, we agree with the reviewer that a different (and better) approach is to include multiple models in the RSA analysis. Therefore, as we described in the response to point #2 of the editor, and to points #2 and #6 of reviewer 1, we now excluded the binary decoding analysis and we went back to the original RSA analyses with multiple models (5 categorical models and 2 low level models).

The results of these analyses are reported in figure 5.

As suggested by the reviewer, the human model is not the only one showing significant correlation with the temporal ROI (also behavioral models and in some cases the pitch model explain part of the functional representation of this ROI). However, the human model is still the only one showing significant correlation with the occipital ROI representation in both groups of blind and not in sighted (see figure 5D). And when we look at the group comparisons of the human model only, in the 2 ROIs (see figure 4B) we still find a significant difference between the 2 groups in both ROIs and a significant interaction group*region, suggesting a shift of the voice representation from the temporal to the occipital cortex in blind subjects.

No other model is showing a significant correlation with the occipital ROI in blind or sighted groups. However, in the temporal cortex we see that also the behavioral model shows a significant correlation in every group. To exclude that the behavioral model was also showing a similar trend as we observed for the human model we, therefore, ran as a supplemental analysis the statistical analyses between groups (see figure 4 – supplemental figure 1). These analyses did not reveal any significant difference between groups and any interaction group*region:

“As a supplemental information, we directly investigated whether there was a statistical difference between groups in the correlation with the behavioral model, both in occipital and in temporal ROIs. (Top panel) RSA results with the behavioral model for the EB / SC groups. The permutation test did not revealed a significant different correlation between EB and SC nor in the occipital ROI (p=0.12), neither in the temporal ROI (p=0.2). Finally, ART analysis 2 Groups X 2 ROIs did not reveal any significant effect of interaction group by region. (Bottom panel) RSA results with the behavioral model for the LB / SC groups. The permutation test did not revealed a significant different correlation between LB and SC nor in the occipital ROI (p=0.6), neither in the temporal ROI (p=0.14). The ART analysis 2 Groups X 2 ROIs did not reveal any significant interaction between groups and regions.”

Reviewer #3:The authors answered my concerns, thank you for the detailed responses. I have only one writing suggestion:– "Studying the same participants" does not necessarily mean "re-analysing data used in our previous work". I would recommend clarifying in the paper that the data from CB and SC participants are the data that were also analysed in the previous paper (i.e., it is a reanalysis).

The reviewer is right, we now describe this point more precisely:

“All the EB and 17 of the SC subjects were the same participants included in Mattioni et al., 2020 and we are re-analysing these data for the current work.”